# Epigenetic regulation of CD38/CD48 by KDM6A mediates NK cell response in multiple myeloma

Jiye Liu [1,14], Lijie Xing[2,14], Jiang Li [3], Kenneth Wen[1], Ning Liu[1,4], Yuntong Liu[1], Gongwei Wu [5], Su Wang [6], Daisuke Ogiya[7], Tian-Yu Song [8,9], Keiji Kurata [1], Johany Penailillo[10], Eugenio Morelli [1], Tingjian Wang[11], Xiaoning Hong [3], Annamaria Gulla[1,12], Yu-Tzu Tai[1], Nikhil Munshi [1], Paul Richardson[1], Ruben Carrasco[10,13], Teru Hideshima[1] & Kenneth C. Anderson [1] ✉

Anti-CD38 monoclonal antibodies like Daratumumab (Dara) are effective in multiple myeloma (MM); however, drug resistance ultimately occurs and the mechanisms behind this are poorly understood. Here, we identify, via two in vitro genome-wide CRISPR screens probing Daratumumab resistance, KDM6A as an important regulator of sensitivity to Daratumumab-mediated antibody-dependent cellular cytotoxicity (ADCC). Loss of *KDM6A* leads to increased levels of H3K27me3 on the promoter of *CD38*, resulting in a marked downregulation in CD38 expression, which may cause resistance to Daratumumab-mediated ADCC. Re-introducing CD38 does not reverse Daratumumab-mediated ADCC fully, which suggests that additional KDM6A targets, including CD48 which is also downregulated upon *KDM6A* loss, contribute to Daratumumab-mediated ADCC. Inhibition of H3K27me3 with an EZH2 inhibitor resulted in CD38 and CD48 upregulation and restored sensitivity to Daratumumab. These findings suggest *KDM6A* loss as a mechanism of Daratumumab resistance and lay down the proof of principle for the therapeutic application of EZH2 inhibitors, one of which is already FDA-approved, in improving MM responsiveness to Daratumumab.

Multiple myeloma (MM) is an incurable hematologic neoplasm characterized by the infiltration of aberrant plasma cells into the bone marrow[1,2]. In the past decade, patient outcomes have improved, in part, due to immunotherapy[3], including monoclonal antibodies (mAb)[4–6], bi-specific T-cell engagers (BiTE)[7–9], antibody-drug conjugates (ADC)[10,11] and chimeric antigen receptor T cells (CAR-T)[12,13]. Specifically, mAbs, which selectively target surface antigens that are highly expressed on MM cells, have achieved remarkable responses, with three mAbs now approved by the U.S. Food and Drug Administration (FDA)[14–16].

[1]Jerome Lipper Multiple Myeloma Center, Lebow Institute for Myeloma Therapeutics, Department of Medical Oncology, Dana-Farber Cancer Institute, Boston, MA 02215, USA. [2]Department of Hematology, Shandong Cancer Hospital and Institute, Shandong First Medical University and Shandong Academy of Medical Sciences, Jinan, Shandong 250117, China. [3]Clinical Big Data Research Center, The Seventh Affiliated Hospital of Sun Yat-Sen University, Shenzhen, Guangdong 518107, China. [4]Department of Marine Bio-Pharmacology, College of Food Science and Technology, Shanghai Ocean University, Shanghai 201306, China. [5]Center for Functional Cancer Epigenetics, Department of Medical Oncology, Dana-Farber Cancer Institute, Boston, MA 02215, USA. [6]Vertex pharmaceuticals, Boston, MA 02210, USA. [7]Department of Hematology and Oncology, School of Medicine, Tokai University, Isehara 259-1193, Japan. [8]Department of Medical Oncology, Dana-Farber Cancer Institute, Boston, MA 02215, USA. [9]Broad Institute of Harvard and MIT, Cambridge, MA 02142, USA. [10]Department of Oncologic Pathology, Dana-Farber Cancer Institute, Boston, MA 02215, USA. [11]Department of Cancer Biology, Dana-Farber Cancer Institute, Boston, MA 02215, USA. [12]Candiolo Cancer Institute, FPO-IRCCS, Candiolo (TO) 10060, Italy. [13]Department of Pathology, Brigham and Women's Hospital, Harvard Medical School, Boston, MA 02215, USA. [14]These authors contributed equally: Jiye Liu, Lijie Xing. ✉e-mail: Kenneth_anderson@dfci.harvard.edu

Daratumumab (Dara) is the first-in-class humanized mAb targeting CD38. It has a high frequency and depth of response in newly diagnosed and relapsed/refractory MM when used alone or in combination with other agents[17–19]. Dara induces MM cell death through different mechanisms, including antibody-dependent cell-mediated cytotoxicity (ADCC), antibody-dependent cellular phagocytosis (ADCP), complement-dependent cytotoxicity (CDC), direct cytotoxicity and immunomodulatory effects. Among these effects, ADCC is the most important mechanism of MM cell killing. The sensitivity of Dara-mediated ADCC is strongly associated with the activity of effector cells[20] like natural killer (NK) cells and with the expression of CD38 on MM cells[21,22]. However, the development of Dara resistance and subsequent relapse is common.

CD38 is a transmembrane glycoprotein that is highly expressed in MM cells and lowly expressed in normal hematological cells including NK cells, B cells, and T cells[23], and its expression on MM cells is correlated with response to Dara treatment[21]. Clinically, CD38 expression on patient MM cells significantly decreases during Dara treatment[21], and low levels of CD38 expression are also observed during MM progression[21]. CD38 expression can be modulated by immunomodulatory drugs (IMiD)[24], all-trans retinoic acid (ATRA)[25], histone deacetylase inhibitors (HDAC)[26], IL-6 from bone marrow stromal cells, and the JAK2 inhibitor ruxolitinib[22]. However, using these agents in the

clinic has not achieved the desired effect, suggesting that the molecular mechanisms underlying CD38 regulation are still not completely understood. In addition, NK cells play a crucial role in mAbs-mediated ADCC, so enhancing the activation of NK cells is an alternative strategy[27].

Here, we perform two genome-wide CRISPR screens, one for genes that regulate Dara-mediated cytotoxicity and one for genes that regulate CD38 expression. We find that *KDM6A* was a hit in both screens and thus investigate its mechanisms, finding that the loss of *KDM6A* increases the level of H3K27me3, resulting in downregulation of both CD38 and CD48 expression, which in turn, leads to reduced ADCC.

## Results

### Genome-wide CRISPR screen for NK-cell-mediated cytotoxicity

To identify the tumor-intrinsic genes involved in resistance to NK-mediated cytotoxicity in the presence of Dara in MM cells, we transduced H929 cells with a pooled genome-scale CRISPR knockout (KO) library, selected with puromycin for 7 days, and exposed them to human primary NK cells at an effector-to-target (E:T) ratio of 1:1 with Dara. This resulted in ~75% tumor cell lysis (Fig. 1a), and the abundance of each sgRNA in the surviving tumor cells was assessed by next-generation sequencing of genomic DNA (Fig. 1b and Supplementary

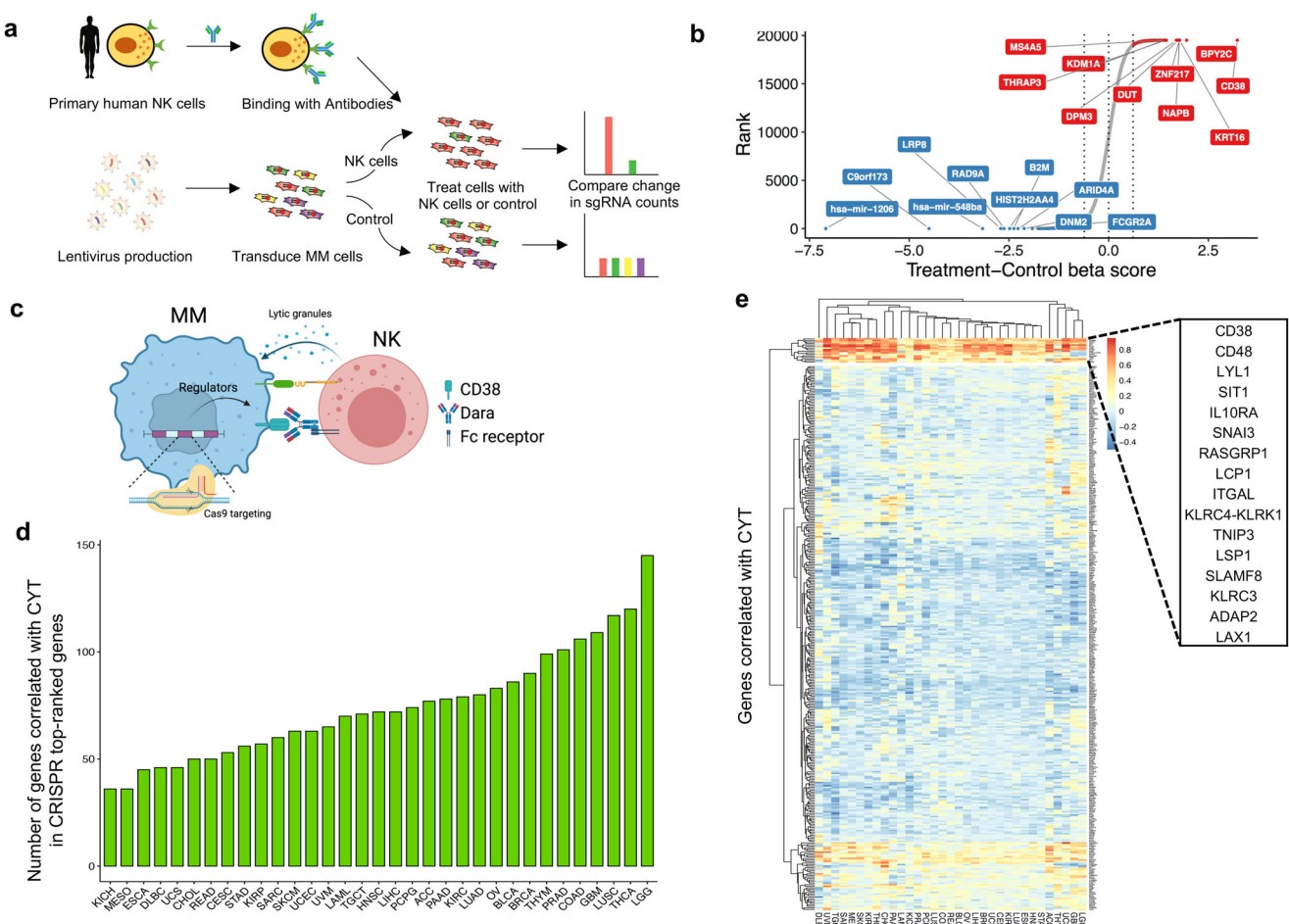

**Fig. 1 | CRISPR screen identifies cell-intrinsic genes regulating the sensitivity of MM cells to Dara-mediated NK cell cytotoxicity. a** Schematic of genome-wide CRISPR screen in a MM cell line treated with Dara and human primary NK cells. **b** Top genes for enriched (red) and depleted (blue) sgRNAs from the screen. Candidate genes were plotted based on the beta score, computed by MaGeCK (Model-based Analysis of Genome-wide CRISPR-Cas9 Knockout) of sgRNAs normalized to control. **c** Schematic of the CRISPR assay to identify essential genes for

Dara-NK-mediated ADCC. **d** TCGA RNA-seq data from 36 human cancer types were analyzed to obtain genes positively correlated with cytolytic (CYT) activity. The number of overlapped genes between our top candidates and CYT activity-related genes was quantified in each cancer type. **e** Heatmap showing the partitioning of the clusters of genes based on Pearson's correlation coefficient values of CRISPR screen hits with CYT activity using pan-cancer TCGA data. Figure 1c was created with BioRender.com.

Data 1), to identify affected genes involved in Dara-mediated ADCC and/or direct NK-cell-mediated cytotoxicity (Fig. 1c). As expected, the sgRNAs targeting *CD38*, the target of Dara, were listed as the top candidate, indicating the screen performed properly.

To identify whether the candidate genes we identified attenuated cytolytic activity in other cancers, we analyzed the gene expression profiles of 11,409 human tumors from 36 different cancer types in the TCGA database (Fig. 1d and Supplementary Data 2) and found a set of 16 genes that are correlated with the cytolytic activity in most of the 36 cancer types (Fig. 1e). Loss of these 16 genes within tumors could play a major role in immune evasion from NK cell-mediated cytotoxicity.

### KDM6A loss leads to CD38 downregulation

We performed another genome-wide CRISPR KO screen to identify the genes regulating CD38 expression in H929 cells (Fig. 2a). After puromycin selection, we sorted cells from the bottom 5% of CD38 expression, meaning the enriched genes positively regulated CD38 (Fig. 2b and Supplementary Data 3). The genes that overlapped in these two screens are implicated in regulating both Dara-mediated ADCC and CD38 expression (Fig. 2c). From the overlapping genes, we were particularly interested in KDM6A, which belongs to the KDM6 family of histone 3 lysine 27 demethylases. It de-represses genes by removing methylation from H3K27 and counteracts the activity of EZH2 of the PRC2 complex, which methylates H3K27. The loss or inactivation of KDM6A occurs in several hematological malignancies including MM[28], and patients with *KDM6A* mutations or deletion have reduced overall survival[29].

To validate whether KDM6A regulates CD38 expression, we generated *KDM6A* KO MM cell lines using different sgRNA sequences and found that loss of *KDM6A* was indeed associated with CD38 downregulation on the protein and mRNA levels (Fig. 2d,e). Moreover, we confirmed that the cell-surface expression of CD38, which is important for Dara binding, was significantly decreased (Supplementary Fig. 1a). For the pool CRISPR KO cell lines, there was still KDM6A expression in the KO cells, so we generated complete KO single clones and found that their CD38 expression was much lower than that in pool KO cells (Fig. 2f).

We next sought to confirm whether the downregulation of CD38 was a specific effect of *KDM6A* KO. We added back *KDM6A* in *KDM6A* KO cells and found that the CD38 level was restored by KDM6A overexpression (Fig. 2g, h and Supplementary Fig. 1b–d). We did not detect a significant difference in CD38 expression between different media conditions, indicating that CD38 was specifically downregulated by *KDM6A* KO in a cell-autonomous manner (Supplementary Fig. 1e, f). These data indicate that loss of *KDM6A* decreased CD38 expression at the transcriptional and cell surface expression levels in myeloma cells.

### KDM6A regulates CD38 expression via H3K27me3 and chromatin accessibility

Given KDM6A's function as an H3K27 demethylase, we hypothesized that KDM6A positively regulated the expression of CD38 by decreasing the level of H3K27me3 on its promoter. We, therefore, performed H3K27me3 ChIP-seq in control and two isogenic *KDM6A* KO cell lines. The *KDM6A*-KO cells had more H3K27me3 loci than control cells (Fig. 3a) and significantly more at the promoter region of *CD38*

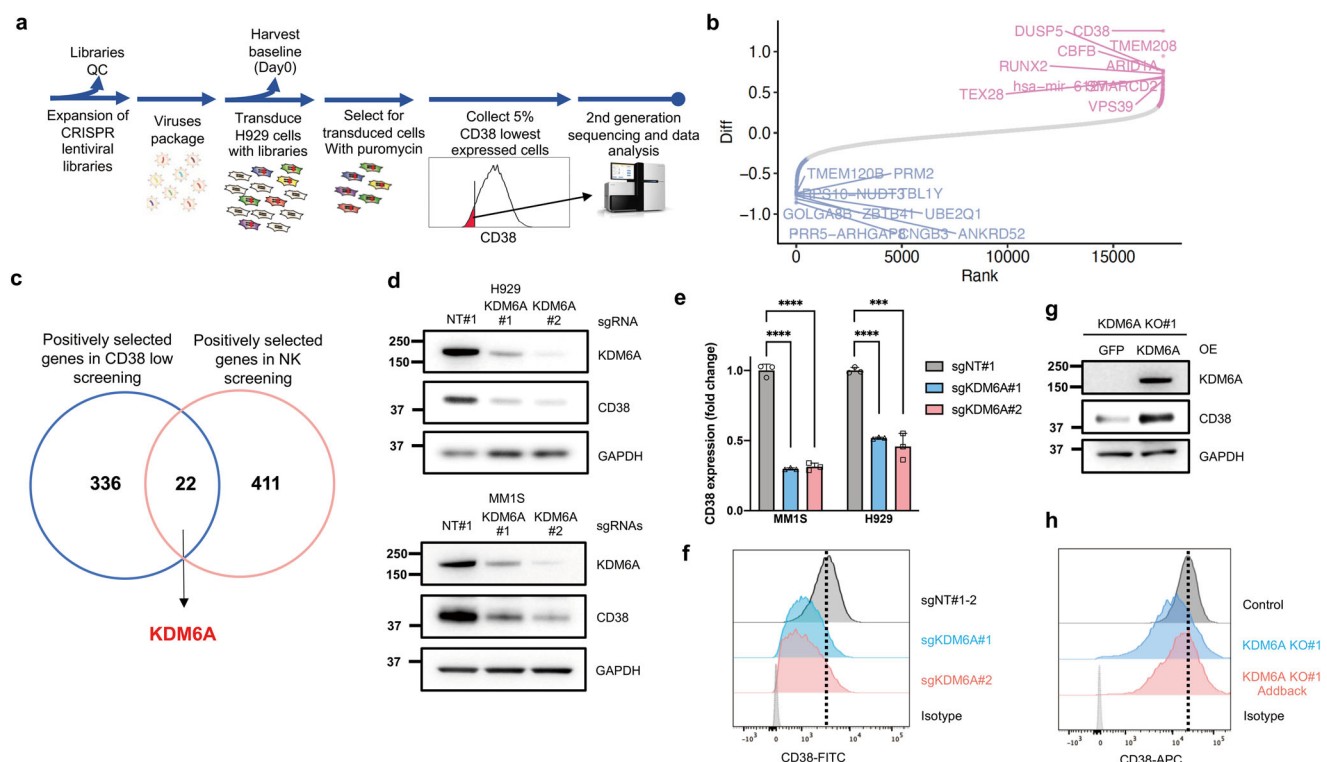

**Fig. 2 | KDM6A loss inhibits the expression of CD38. a** Schematic of the second genome-wide CRISPR screen in a MM cell line. Five percent of cells with the lowest expression of CD38 were collected for next-generation sequencing. **b** Top genes for enriched (red) and depleted (blue) gRNAs from the screen. **c** Venn diagram showing the overlapped enriched genes between the two CRISPR screens. **d** Western blot of CD38 protein levels in MM cell lines with CRISPR-mediated knock (KO) of *KDM6A*. **e** q-RT-PCR for CD38 mRNA in MM cell lines transfected with indicated sgRNAs. Data were normalized against GAPDH (mean ± SEM, *n* = 3 biologically independent

experiments). ***$p < 0.001$; ****$p < 0.0001$ (two-sided student's *t* test). **f** Representative flow cytometry analysis of CD38 expression in H929 single clones transfected with indicated sgRNAs. **g** Western blotting of CD38 protein levels after ectopic overexpression of KDM6A in *KDM6A*-KO H929 cells. OE, overexpression. Three independent experiments were performed and similar results were obtained. **h** Representative flow cytometry analysis of CD38 expression level after ectopic overexpression of KDM6A in *KDM6A*-KO H929 cells. Source data are provided as a Source Data file.

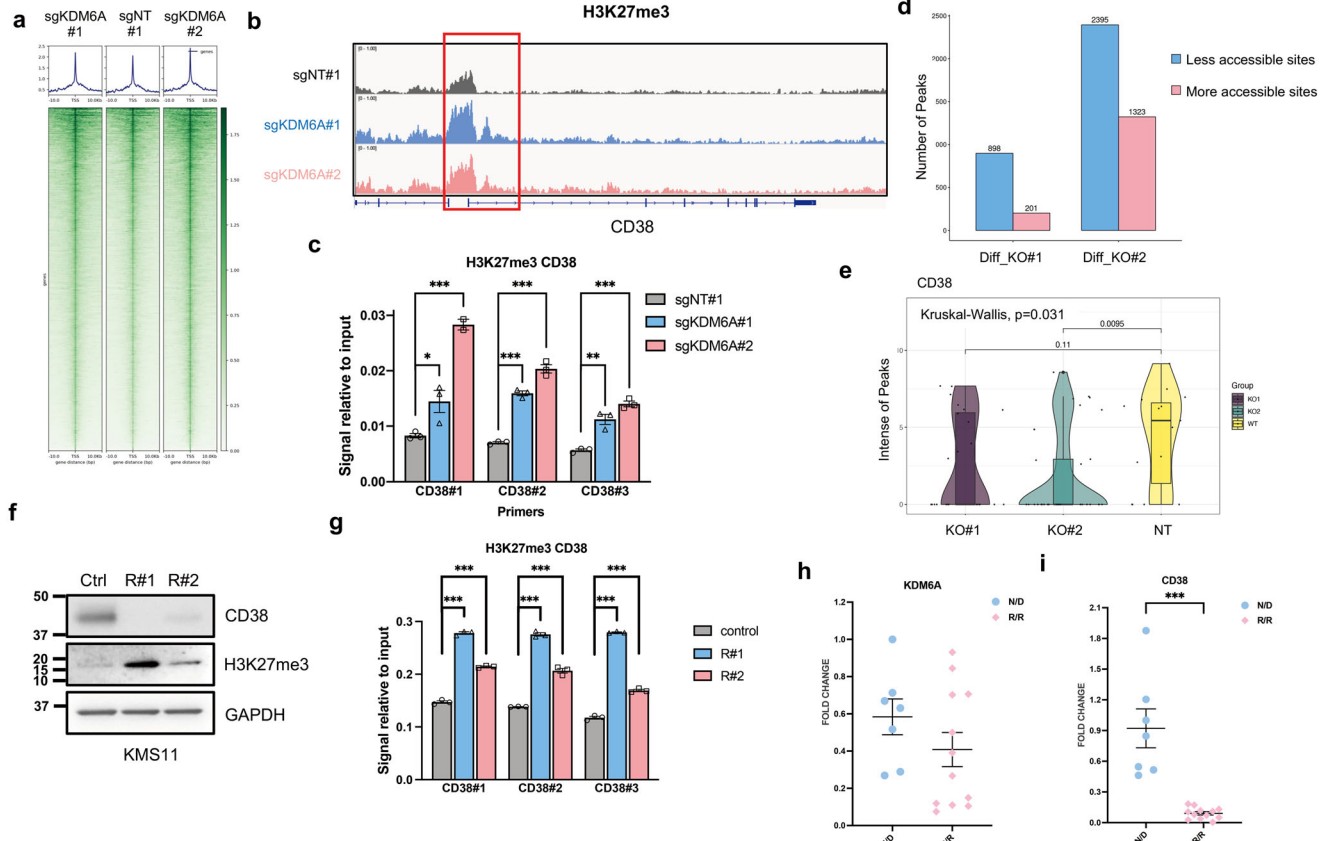

**Fig. 3 | KDM6A regulates CD38 expression at the transcriptional level.**
**a** Genome-wide heatmaps of H3K27me3 ChIP-seq peak centered signal in *KDM6A* WT and KO H929 cells. **b** ChIP-seq density profiles for H3K27me3 at the *CD38* promoter region in *KDM6A* WT and KO H929 cells. **c**, H3K27me3 ChIP-qPCR analysis at the *CD38* gene in *KDM6A* WT and KO H929 cells (mean ± SEM, *n* = 3 biologically independent experiments). *$p < 0.05$; **$p < 0.01$; ***$p < 0.001$ (two-sided student's *t* test). **d** Genome-wide analysis of differentially accessible chromatin sites (|log2 fold change|> 0.5) following *KDM6A* KO in H929 cells. **e** The intensities of peaks at the *CD38* locus across the three samples are compared. The violin plot describes the distribution of peak intensity values. The width describes how often the intensity value occurs in the data set. The ends of the middle vertical line define the

minimum and maximum values. Box plots represent the median, 25th, and 75th percentiles, and the thick dots represent outliers. The actual intensity values under different conditions are also marked in the figure. **f** Western blot of CD38 and H3K27me3 levels in KMS11 Dara-sensitive and Dara-resistant cells. Three independent experiments were performed and similar results were obtained. **g** H3K27me3 ChIP-qPCR analysis at the *CD38* gene in KMS11 Dara-sensitive and Dara-resistant cells (mean ± SEM, *n* = 3 biologically independent experiments). ***$p < 0.001$ (two-sided student's *t* test). q-RT-PCR for *KDM6A* (**h**) and *CD38* (**i**) mRNA in newly diagnosed (N/D) (*n* = 7) and Dara-resistant (R/R) (*n* = 12) patient MM samples. Data were normalized against GAPDH (mean ± SEM). ***$p < 0.001$ (two-sided student's *t* test). Source data are provided as a Source Data file.

(Fig. 3b), evidenced by ChIP-qPCR with three different primers (Fig. 3c). Conversely, the re-introduction of *KDM6A* into *KDM6A*-KO cells decreased the H3K27me3 level at the promoter region of *CD38* (Supplementary Fig. 1g). Interestingly, the H3K27Ac level, which is recognized as a marker of active enhancers[30], was increased at the same time (Supplementary Fig. 1h).

KDM6A facilitates gene expression by removing a suppressive chromatin modification and increasing chromatin accessibility. We, therefore, performed ATAC-seq to check chromatin accessibility and found a substantially large number of differentially accessible sites between control and *KDM6A*-KO cells (Supplementary Data 4). There were 898 and 2395 less-accessible sites in two different *KDM6A*-KO cell lines relative to the control, respectively (Fig. 3d). In particular, the peak on the *CD38* gene had decreased intensity, leading to the inhibition of *CD38* transcription (Fig. 3e).

To explore the mechanism of acquired resistance in MM cells, we generated a Dara-resistant MM cell line by continuously exposing MM cells to NK and Dara for several rounds (Supplementary Fig. 2a). In these resistant cells, CD38 expression was significantly decreased (Fig. 3f and Supplementary Fig. 2b), leading to attenuated sensitivity to ADCC (Supplementary Fig. 2c). Importantly, the CD38 promoter had

increased levels of H3K27me3 and decreased levels of H3K27Ac (Fig. 3g and Supplementary Fig. 2d).

**KDM6A KO renders MM cells resistant to Dara-mediated ADCC**
Importantly, we found that *KDM6A* and *CD38* mRNA expression was decreased in Dara-resistant patient samples relative to newly diagnosed patients (Fig. 3h, i). To evaluate whether the decreased CD38 expression caused by *KDM6A* KO affects Dara-mediated ADCC, we co-incubated MM cells with Dara and human primary NK cells or PBMC cells. The parental MM cells were profoundly lysed, whereas *KDM6A*-KO cells were not (Fig. 4a and Supplementary Fig. 3a,b). This also occurred with another anti-CD38 mAb, Isatuximab, indicating that ADCC resistance was specific to CD38 expression (Fig. 4b). Importantly, when we re-introduced CD38 into *KDM6A*-KO cells, their sensitivity to Dara-mediated ADCC was partially restored (Supplementary Fig. 3c,d). Subsequently, we tested the sensitivity of Dara-mediated ADCC in a *KDM6A*-null xenograft model. After tumor growth, we gave the mice an i.v. injection of human primary NK cells and i.p. injection of Dara. The *KDM6A*-KO engrafted tumors were significantly resistant to this treatment (Fig. 4c and Supplementary Fig. 3e), associated with shorter survival times (Fig. 4d).

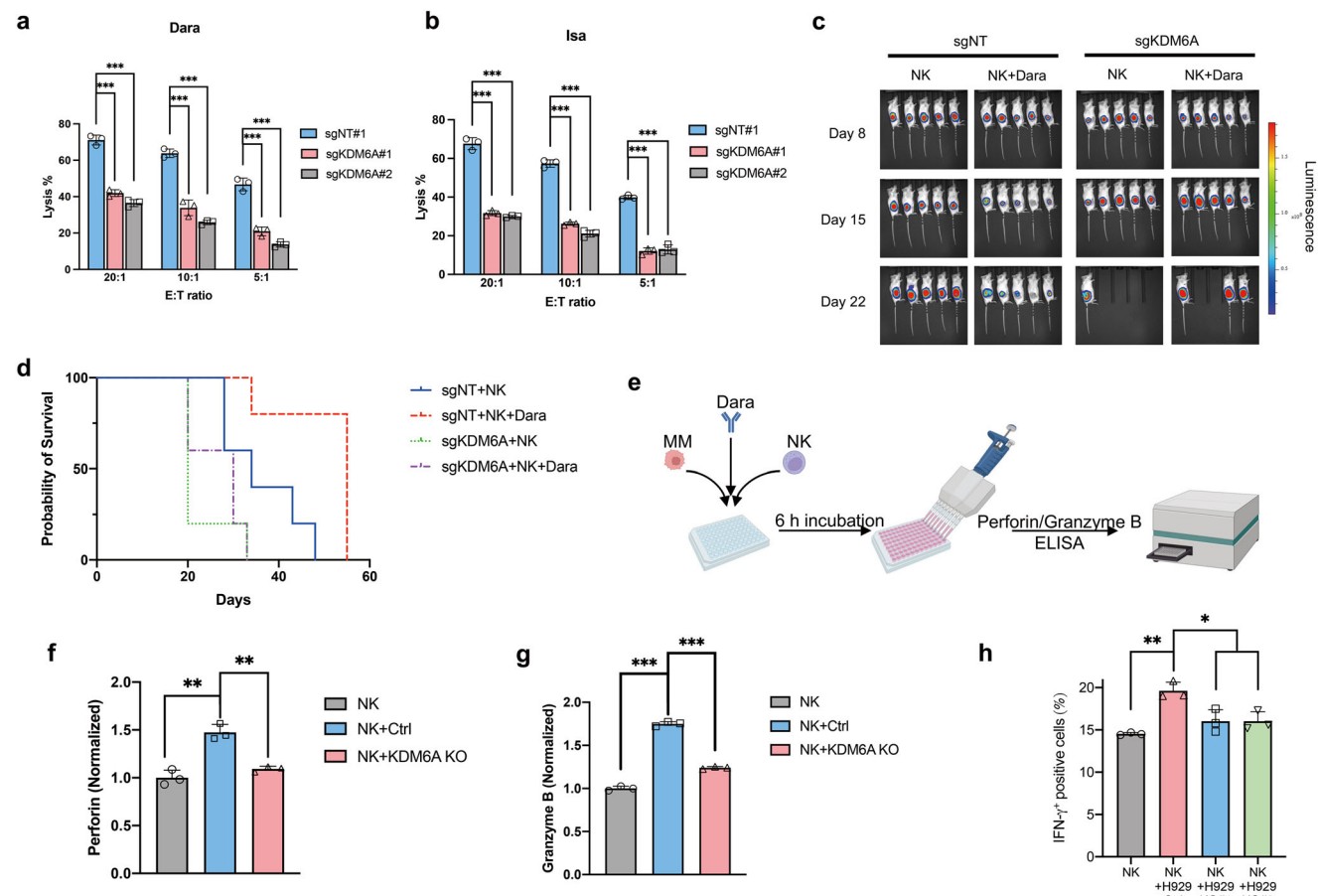

**Fig. 4 | KDM6A KO decreases CD38 mAb-mediated ADCC.** *KDM6A* WT and KO H929 cells were co-cultured with primary human NK cells and Dara (**a**) or Isa (**b**), and subjected to ADCC assay (mean ± SEM, *n* = 3 biologically independent experiments). ***$p < 0.001$ (two-sided student's *t* test). **c** Bioluminescent imaging of mice transplanted with *KDM6A* KO or WT H929 cells and treated with human primary NK cells or NK+Dara (8 mg/kg). Representative images of five mice for each group are shown at the indicated time. **d** Kaplan-Meier survival curves of mice in **c**. Schematic design of ELISA assay (**e**). *KDM6A* WT or KO H929 cells were co-cultured with Dara or primary NK cells for 6 hours, and the supernatant was collected for Perforin (**f**) and Granzyme B (**g**) ELISA assay (mean ± SEM, *n* = 3 biologically independent experiments). **$p < 0.01$; ***$p < 0.001$ (Student's test). **h** Intracellular IFN-γ staining of primary NK cells co-cultured with *KDM6A* WT or KO H929 cells and Dara for 6 hours (mean ± SEM, *n* = 3 biologically independent experiments). *$p < 0.05$; **$p < 0.01$ (two-sided student's *t* test). Source data are provided as a Source Data file.

NK cells kill cancer cells through the secretion of the pore-forming protein perforin and granzymes, we next examined whether the resistance of *KDM6A*-KO cells to ADCC was due to attenuated NK activity. We co-incubated MM and primary NK cells with Dara, then collected culture supernatant for an ELISA detecting perforin and granzyme B (Fig. 4e), and found that *KDM6A* KO suppressed NK activity by lowering the secretion of perforin and granzyme B (Fig. 4f, g). We also found that human NK cells produced lower amounts of IFN-γ when co-cultured with *KDM6A*-KO MM cells than with control MM cells in the presence of Dara (Fig. 4h). Taken together, these data demonstrate that the loss of *KDM6A* mediates resistance of MM cells to Dara-mediated ADCC through CD38 downregulation and suggests that the KDM6A expression of MM cells regulates the activity of NK cells.

## KDM6A regulates CD48 expression in MM cells

Overexpressing CD38 in *KDM6A*-KO MM cells only partially rescued Dara-mediated ADCC (Supplementary Fig. 3c, d), even when we utilized an anti-SLAMF7 mAb, elotuzumab, *KDM6A* KO MM cells continued to exhibit resistance to ADCC (Supplementary Fig. 4a, b). These findings imply the presence of an additional mechanism mediating ADCC resistance by KDM6A beyond the regulation of CD38 expression. Moreover, we found that *KDM6A* KO MM cells efficiently evade direct primary NK cell and NK cell lines-mediated cytotoxicity (Fig. 5a and Supplementary Fig. 4c–e). The IFN-γ produced by primary NK cells

was inhibited when co-cultured with *KDM6A*-KO cells (Fig. 5b). These data imply that KDM6A may function as a regulator of NK activity in MM cells. To investigate how KDM6A regulates NK activity, we performed RNA-seq analysis in *KDM6A*-KO cells (Fig. 5c and Supplementary Data 5). As expected, gene enrichment analysis revealed that PRC2-regulated genes were highly repressed in *KDM6A*-KO cells (Supplementary Fig. 4f). Of note, CD48 was highly repressed and was also a top hit in the previous CRISPR screens (Fig. 5d).

CD48 is a ligand expressed on cancer cells that binds with its receptor 2B4 on NK cells for their activation. In *KDM6A*-KO cells, CD48 had significantly reduced mRNA and surface expression (Fig. 5e, f) and a higher level of H3K27me3 on its gene (Fig. 5g, h), which corresponded to profoundly reduced chromatin accessibility in the *CD48* promoter region (Fig. 5i). We also validated other well-studied NK activating and inhibitory ligands and found there were no significant alterations in the expression of other evaluated NK cell ligands (Supplementary Fig. 4g, h). Finally, we found that CD48 expression levels in Dara-resistant patient MM samples are lower than in the newly diagnosed MM (Fig. 5j). In addition, using the TCGA cancer RNA-seq dataset and TIMER[31], we found that the expression of CD48 and KDM6A was positively correlated with intertumoral NK cell abundance in many human cancer types, suggesting that KDM6A and CD48 affect NK-mediated tumor immunity in a variety of human cancers (Supplementary Fig. 4i, j).

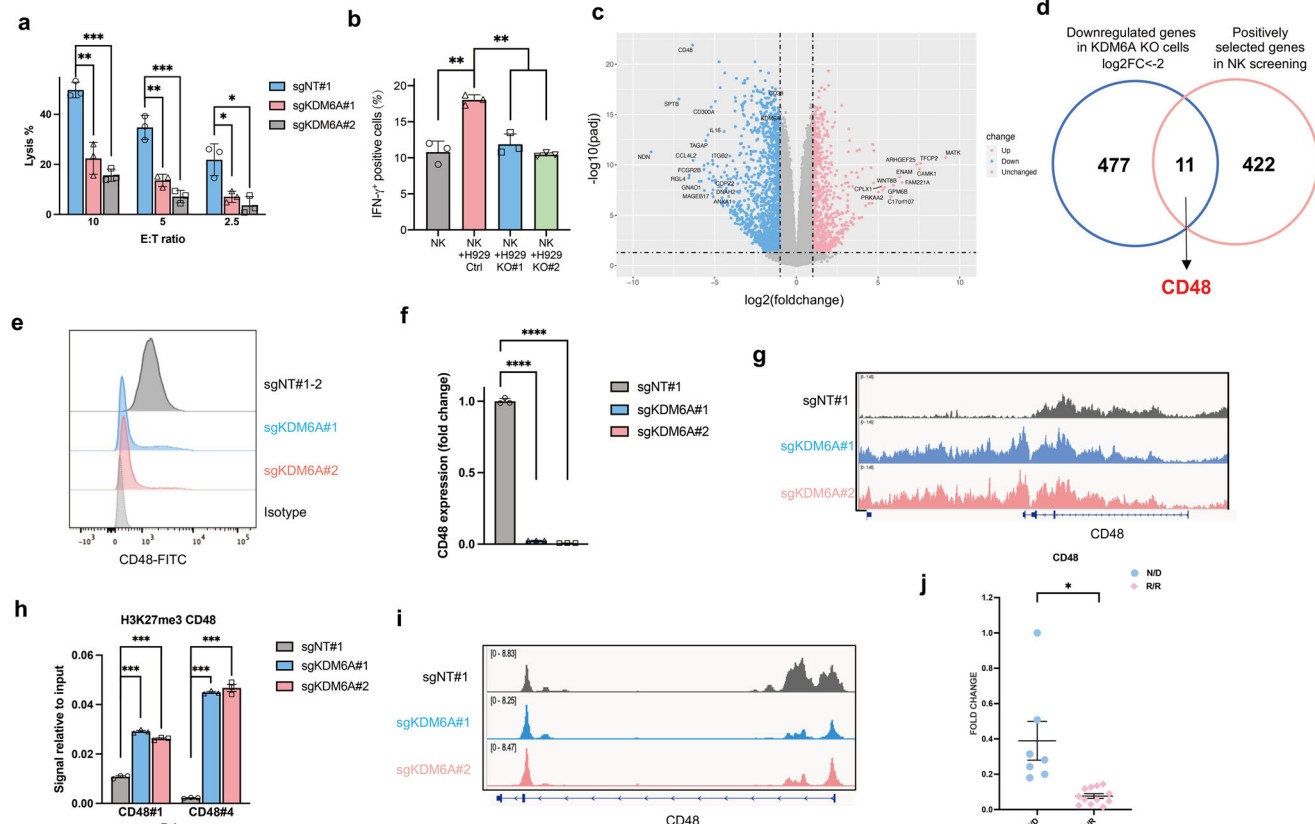

**Fig. 5 | KDM6A regulates CD48 expression in MM cells. a** Normalized lysis percentage of *KDM6A* WT or KO H929 cells after co-culture with primary NK cells at different E:T ratios with IL-2 for 6 hours (mean ± SEM, *n* = 3 biologically independent experiments). **p* < 0.05; ***p* < 0.01; ****p* < 0.001 (two-sided student's *t* test). **b** Intracellular IFN-γ staining of primary NK cells co-cultured with *KDM6A* WT or KO H929 cells with IL-2 for 6 hours (mean ± SEM, *n* = 3 biologically independent experiments). ***p* < 0.01 (two-sided student's *t* test). **c** Volcano plot of differentially expressed genes in *KDM6A* KO H929 cells compared with WT cells assessed by RNA-seq. **d** Venn diagram showing the overlapped enriched genes between the CRISPR screen and RNA-seq. **e**, Representative flow cytometry analysis of CD48 expression in H929 transfected with indicated sgRNAs. **f** q-RT-PCR for CD48 mRNA in H929

cells transfected with indicated sgRNAs. Data were normalized against GAPDH (mean ± SEM, *n* = 3 biologically independent experiments). *****p* < 0.0001 (two-sided student's *t* test). **g**, ChIP-seq density profiles for H3K27me3 on the *CD48* gene in *KDM6A* WT and KO H929 cells. **h**, H3K27me3 ChIP-qPCR analysis at the *CD48* gene in *KDM6A* WT and KO H929 cells (mean ± SEM, *n* = 3 biologically independent experiments). ****p* < 0.001 (two-sided student's *t* test). **i**, ATAC-seq density profiles at the *CD48* promoter region in *KDM6A* WT and KO H929 cells. **j**, q-RT-PCR for CD48 mRNA in newly diagnosed (N/D) (*n* = 7) and Dara-resistant (R/R) (*n* = 12) patient samples. Data were normalized against GAPDH (mean ± SEM). **p* < 0.05 (two-sided student's *t* test). Source data are provided as a Source Data file.

## CD48 mediates ADCC sensitivity by regulating NK activity

To establish whether CD48 loss downregulates NK activity leading to ADCC resistance, we generated *CD48*-KO cells using CRISPR, and confirmed the depletion of cell surface expression of CD48 in these cells (Fig. 6a). Upon exposure to primary NK cells with or without Dara, the lysis of *CD48*-KO MM cells was profoundly inhibited (Fig. 6b,c and Supplementary Fig. 5a). The effects were similar to those observed in *KDM6A*-KO cells (Fig. 5a and Supplementary Fig. 5b). When we over-expressed CD48 in *KDM6A*-KO cells (Fig. 6d), we found these cells significantly increased the IFN-γ expression in primary NK cell fractions (Supplementary Fig. 5c, d), as well as secretion of perforin and granzyme B from NK cells (Supplementary Fig. 5e, f). Importantly, CD48 overexpression significantly restored the sensitivity of *KDM6A*-KO MM cells to Dara-mediated ADCC (Fig. 6e and Supplementary Fig. 5g) and recovered IFN-γ production, secretion of perforin and granzyme B by NK cells (Fig. 6f-h). Taken together, these data suggest that KDM6A mediates Dara-mediated ADCC sensitivity through both CD38 upregulation and CD48 upregulation, with associated increased NK activity.

## EZH2 inhibitors increase CD38 and CD48 expression and enhance Dara-mediated ADCC

KDM6A acts to oppose the EZH2/PRC2 complex by demethylating H3K27me3; conversely, inactivating KDM6A leads to increased

H3K27me3 and inhibition of gene transcription (Fig. 7a). Several pre-clinical studies have shown that EZH2 can be a therapeutic target in MM. Therefore, we hypothesized that inhibiting EZH2 may restore the balance of gene expression by downregulating H3K27me3, resulting in the upregulation of CD38 and CD48 expression in *KDM6A*-KO cells. Taze-metostat (Taze), an FDA-approved EZH2 inhibitor (EZHi), has already provided meaningful and sustained responses for relapsed or refractory follicular lymphoma patients[32]. We found that Taze increased the expression of CD38 and CD48 at the protein (Fig. 7b), mRNA (Fig. 7c, d), and surface expression levels (Fig. 7e), especially in *KDM6A*-KO cells. The expression of CD48 did not increase as much as CD38, suggesting other regulatory mechanisms. For example, KDM6A also interacts with p300, H3K4 methyltransferases, and the SWI/SNF complex, besides functioning as a demethylase. ChIP-Q-PCR data showed that the H3K27me3 level on the *CD38* and *CD48* genes was reduced by Taze in MM cell lines (Fig. 7f,g). Importantly, we validated the above findings with another EZH2 inhibitor, GSK343, which inhibits EZH2 catalytic activity (Supplementary Fig. 6a–g)[33]. Interestingly, we found that Taze did not change the CD38 expression level in NK cells (Supplementary Fig. 6h).

Finally, we sought to determine whether Taze can re-sensitize the *KDM6A*-KO cells to Dara-mediated ADCC. We treated MM cells with Taze for 4 days, then co-incubated them with Dara or Isa and human primary NK cells or PBMC. The sensitivity of the *KDM6A*-KO cells to

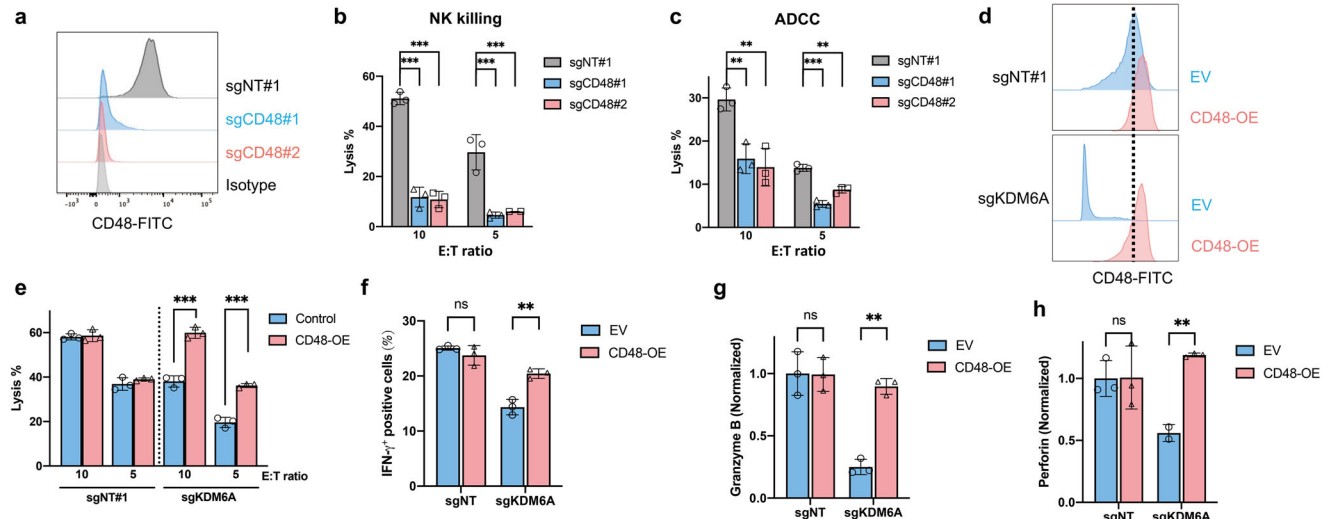

**Fig. 6 | CD48 mediates ADCC sensitivity through NK activity regulation.**
**a** Representative flow cytometry analysis of CD48 expression in H929 cells transfected with indicated sgRNAs. **b** Normalized lysis percentage of *CD48* WT or KO H929 cells after co-culture with primary NK cells at different E:T ratios with IL-2 for 6 hours (mean ± SEM, *n* = 3 biologically independent experiments). ***$p < 0.001$ (two-sided student's *t* test). **c** *CD48* WT and KO H929 cells were co-cultured with primary human NK cells and Dara at different E:T ratios, and subjected to ADCC assay (mean ± SEM, *n* = 3 biologically independent experiments). **$p < 0.01$; ***$p < 0.001$ (two-sided student's *t* test). **d** Representative flow cytometry analysis of CD48 expression in *KDM6A* KO and control H929 cells after ectopic overexpression of CD48. **e** ADCC assay of *KDM6A* KO and WT H929 cells after ectopic

overexpression of CD48 and co-cultured with primary human NK cells and Dara at different E:T ratios (mean ± SEM, *n* = 3 biologically independent experiments). ***$p < 0.001$ (two-sided student's *t* test). **f** Intracellular IFN-γ staining of primary NK cells co-cultured with *KDM6A* WT or KO H929 cells after ectopic overexpression of CD48 with Dara for 6 hours (mean ± SEM, *n* = 3 biologically independent experiments). ns, not significant; **$p < 0.01$ (two-sided student's *t* test). *KDM6A* KO or control H929 cells after ectopic overexpression of CD48 were co-cultured with Dara and primary NK cells for 6 hours, and the supernatant was collected for granzyme B (**g**) and perforin (**h**) ELISA assay (mean ± SEM, *n* = 3 biologically independent experiments). ns, not significant; **$p < 0.01$ (two-sided student's *t* test). Source data are provided as a Source Data file.

ADCC was profoundly restored by Taze treatment (Fig. 7h, i and Supplementary Fig. 7a, b). In addition, NK cell activity was also restored, evidenced by increased granzyme B secretion (Fig. 7j). We also confirmed that Taze increased Dara-mediated killing of MM patient cells (Fig. 7k) and increased CD38 and CD48 expression (Supplementary Fig. 7c, d). Even in a CD38 low-expression cell line, Taze increased CD38 expression and enhanced ADCC activity (Supplementary Fig. 7e–g). These data suggest that adding an EZH2 inhibitor to the therapeutic regimen would overcome Dara resistance by upregulating CD38 and NK cell activity.

## Discussion

KDM6A, also known as UTX, is a histone demethylase that removes the trimethylation of H3K27 (H3K27me3), which is catalyzed by the EZH2-containing polycomb repressive complex 2 (PRC2)[34]. Loss or inactivation of KDM6A occurs in multiple human cancers, including MM[28,29]. In many cancers, KDM6A functions as a tumor suppressor. Nevertheless, overexpression of KDM6A in breast cancer increases the proliferation of cancer cells, suggesting that the function of KDM6A might be tissue-specific[35] or that the genes targeted by KDM6A are different among cell types[36]. In MM, KDM6A loss leads to an enhanced malignant phenotype and sensitizes the cell to EZH2 inhibition[37]. However, little is known about the role of KDM6A in MM immunotherapy. Here, we describe a new mechanism of KDM6A, whereby its loss induces Dara resistance by downregulating CD38 and CD48 on MM cells, thereby decreasing NK cell activity.

Previous studies have shown that different variants of FcγRs expressed on NK cells are associated with anti-tumor activity of Dara[20]. In addition, Dara can also mediate NK fratricide by binding to CD38 on the NK cell surface[38]. Deletion of CD38 in human primary NK cells eliminates Dara-mediated fratricide and enhances the anti-tumor activity of NK cells[39]. In recent years, many CRISPR screens have been done to explore the mechanism of tumor cell resistance to NK cell treatment. Different mechanisms of resistance were found in

different approaches and in different cancer types[40–42]. In our study, we focused on the screens for mAb-mediated ADCC, and the mechanisms underlying NK cell resistance identified in our study are distinct from those findings. The diversity of identified mediators emphasizes the complexity of tumor cell resistance mechanisms against immune cells. Further research and comparative analyses across different mediators are crucial for a comprehensive understanding of the regulatory networks governing immune evasion in tumor cells. This could potentially uncover new therapeutic targets and strategies for overcoming resistance in cancer immunotherapy. In this study, we focused on the intra-tumor factors that affect NK cell activity. We found that KDM6A epigenetically upregulates the expression of CD48 in MM cells, whereas loss of CD48 has a known role in the evasion of NK cell-mediated surveillance in hematological neoplasms[43,44].

In recent years, various drugs have been developed to target epigenetic regulators in the treatment of cancers, such as Azacitidine targeting DNMT1 and Panobinostat targeting pan-histone deacetylase[45]. Tazemetostat is a potent selective EZH2 inhibitor that was approved by the FDA in 2020 for the treatment of epithelioid sarcoma[46] and follicular lymphoma. Several preclinical studies have found that EZH2 inhibitors have potent anti-MM activity as a monotherapy or in combination with other conventional drugs[47]. In our current study, we found that the EZH2 inhibitor can overcome the resistance of MM cells to Dara-induced ADCC triggered by the loss of KDM6A through upregulating CD38 and CD48 expression. At the same time, the EZH2 inhibitor does not increase the CD38 expression level in NK cells, suggesting it will not enhance Dara-mediated NK fratricide.

In summary, our data reveal a molecular mechanism whereby KDM6A mediates ADCC not only through regulating CD38 but also by modulating NK activity through CD48 regulation, demonstrating the therapeutic potential of EZH2 inhibitor in MM treatment to overcome Dara resistance and providing the preclinical rationale for a Dara-based combination therapeutic strategy to improve patient outcome in MM (Fig. 8).

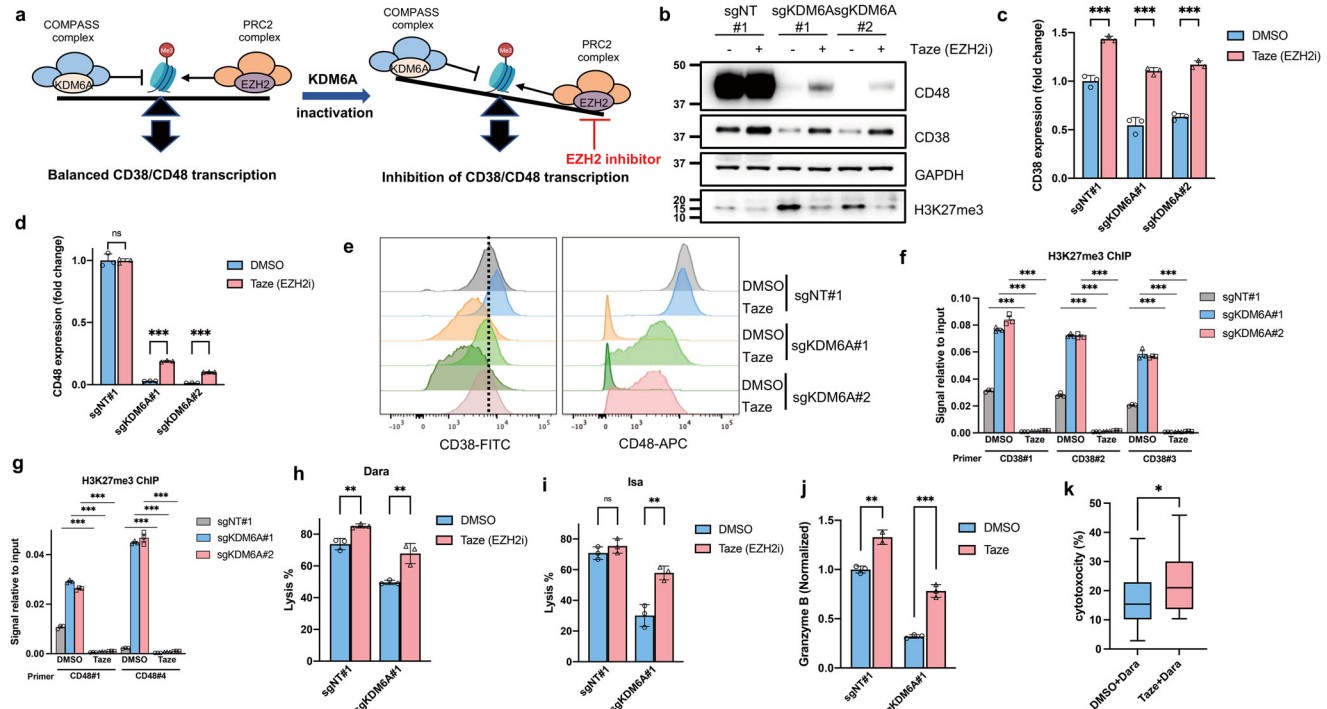

**Fig. 7 | EZH2 inhibitors increase CD38 and CD48 expression and enhance Dara-mediated ADCC. a** Schematic utilizing EZH2 inhibitor to balance the expression of CD38 and CD48 in *KDM6A*-KO MM cells. **b**, Western blot of CD38 and CD48 protein levels in *KDM6A*-KO and control H929 cells treated with Taze (5 μM) for 4 days. q-RT-PCR for CD38 (**c**) and CD48 (**d**) mRNA in *KDM6A*-KO and control H929 cells treated with Taze (5 μM) for 4 days. Data were normalized against GAPDH (mean ± SEM, $n = 3$ biologically independent experiments). ns, not significant; ***$p < 0.001$ (two-sided student's *t* test). **e** Representative flow cytometry analysis of CD38 and CD48 expression in *KDM6A* WT and KO H929 cells treated with Taze (5 μM) for 4 days. H3K27me3 ChIP-qPCR analysis at the *CD38* (**f**) and *CD48* (**g**) genes in *KDM6A* WT and KO H929 cells (mean ± SEM, $n = 3$ biologically independent experiments). ***$p < 0.001$ (two-sided student's *t* test). *KDM6A* WT and KO H929 cells were treated with Taze (5 μM) for 4 days, then co-cultured with primary human

NK cells and Dara (**h**) or Isa (**i**) and subjected to ADCC assay (mean ± SEM, $n = 3$ biologically independent experiments). ns, not significant; **$p < 0.01$ (two-sided student's *t* test). **j** *KDM6A* WT and KO H929 cells were treated with Taze (5 μM) for 4 days, and then co-cultured with Dara and primary NK cells for 6 hours. The supernatant was collected for granzyme B ELISA assay (mean ± SEM, $n = 3$ biologically independent experiments). **$p < 0.01$; ***$p < 0.001$ (two-sided student's *t* test). **k** The *p*atients' MM cells were treated with DMSO or Taze (5 μM) for 4 days, followed by the addition of Dara and incubation for 12 hours. Cell cytotoxicity was assessed by flow cytometry. ($n = 10$ for each group). Box plots represent the median, 25th, and 75th percentiles, and whiskers represent the values of min and max. *$p < 0.05$ (two-sided paired student's *t* test). Source data are provided as a Source Data file.

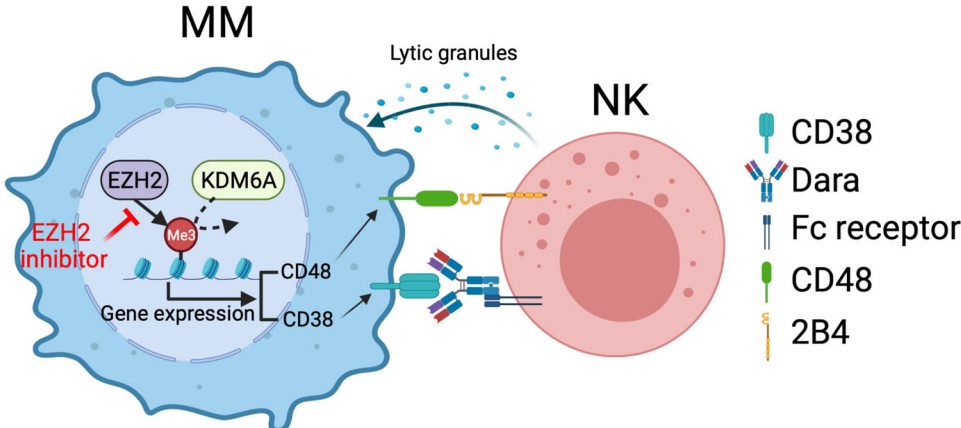

**Fig. 8 | Schema of epigenetic regulation of NK cell-mediated immune response in multiple myeloma monoclonal antibody therapy.** The loss or inactivation of *KDM6A* increased the level of H3K27me3, resulting in the downregulation of both CD38 and CD48 expression, which led to reduced ADCC. Lowering the H3K27me3 with an EZH2 inhibitor restored sensitivity to Dara through CD38 and CD48 upregulation. This Figure was created with BioRender.com.

## Methods

The research methods applied in this study followed the guidelines of the World Medical Association's Declaration of Helsinki and subsequent revisions, and the study was reviewed and approved by the ethics committee of Dana-Farber Cancer Institute.

## Cell lines

H929, MM1.S, U266, and human embryonic kidney (HEK) 293 T cells were purchased from the American Type Culture Collection (ATCC). KMS-11 cells were purchased from Deutsche Sammlung von Mikroorganismen und Zellkulturen (DSMZ). All cell lines were verified by short

tandem repeat (STR) DNA fingerprinting analysis (Molecular Diagnostic Laboratory, DFCI) and tested negative for mycoplasma using the MycoAlert Mycoplasma Detection Kit (Lonza). All cells were grown at 37 °C in 5% $CO_2$. MM.1 S, H929, and U266 cells were maintained in RPMI 1640 medium; HEK293T cells were maintained in Dulbecco's modified Eagle's medium (DMEM). All media were supplemented with 10% fetal bovine serum (FBS), 1X antibiotic-antimycotic, 1X GlutaMAX, and 1X Hepes.

### Primary MM cells

Primary MM cells were obtained from bone marrow samples of patients after informed consent and approved by the Institutional Review Board of the Dana-Farber Cancer Institute. Mononuclear cells were separated by using Ficoll-Paque PLUS (Cytiva). Primary MM cells were purified by CD138 positive selection with anti-human CD138 Microbeads (Miltenyi).

### Reagents and antibodies

Daratumumab and Isatuximab were purchased from the Department of Pharmacy at Dana-Farber Cancer Institute; Tazemetostat and Elotuzumab were purchased from Selleckchem; GSK343 was purchased from Cell Signaling Technology; Tazemetostat and GSK343 were dissolved in dimethyl sulfoxide (DMSO) and stored at −20 °C for up to 6 months. For all cell-based experiments, drugs were diluted at least by 1:1000 to ensure that the final DMSO concentration was lower than 0.1%. Human recombinant IL-2 was purchased from R&D Systems. Antibodies were obtained as follows: KDM6A (no. 33510, Cell Signaling Technology, 1:1000), CD38 (no. 51000, Cell Signaling Technology, 1:1000), CD48 (no. 29499, Cell Signaling Technology, 1:1000), GAPDH (no. 5174, Cell Signaling Technology, 1:1000), H3K27me3 (no. 9733, Cell Signaling Technology, 1:1000), anti-rabbit immunoglobulin G (IgG), horseradish peroxidase (HRP)-linked Ab (no. 7074, Cell Signaling Technology, 1:2000), FITC anti-human CD38 (no. 356610, Biolegend, 1:20), APC anti-human CD38 (no. 356606, Biolegend, 1:20), FITC mouse IgG1 (no. 400110, Biolegend, 1:20), APC mouse IgG1 (no. 981806, Biolegend, 1:20), APC anti-human CD138 (no. 356506, Biolegend, 1:20), FITC anti-human IFN-γ (no. 502506, Biolegend, 1:20), APC anti-human CD48 (no. 336714, Biolegend, 1:20), Alexa Fluor 488 anti-human MICA/B (no. 320912, Biolegend, 1:20), APC anti-human CD56 (no. 985906, Biolegend, 1:20), APC anti-human HLA-A,B,C (no. 311409, Biolegend, 1:20), APC anti-human CD253 (no. 308209, Biolegend, 1:20), FITC anti-human CD319 (no. 331817, Biolegend, 1:20), FITC anti-human CD155 (no. 337627, Biolegend, 1:20), PE anti-human ULBP-2/5/6 (no. FAB1298P, R&D systems, 1:20), BV421 anti-human CD178 (no. 306411, Biolegend, 1:20).

### Two cell type genome-wide CRISPR-Cas9 screen

The human GeCKOv2 sgRNA library was purchased from Addgene and amplified according to Dr. Zhang's lab protocol[48]. To produce lentivirus, HEK293T cells were plated in T150 flasks one day before transfection at 70% confluency. Two hours before transfection, DMEM media was replaced with 12 ml serum-free OptiMEM media. For each flask, 20 μg of GeCKOv2 A library plasmid, 15 μg psPAX2 (Addgene #12260), 10 μg pMD2.G (Addgene #12259), and 200 μl PLUS reagent diluted in 4 ml Opti-MEM were combined with 150 μL Lipofectamine 2000 diluted in 4 ml Opti-MEM. The transfection mixture was left for 20 min and then added dropwise to the cells. 6 h after transfection, the media was replaced by 20 ml fresh complete DMEM media for each flask. Virus-containing media was collected 72 h post-transfection followed by centrifugation at 800 x g for 10 min to pellet cell debris. Filtration was then performed with a 0.45 μm low protein binding membrane (Millipore #SE1M003M00). Viral supernatants were concentrated by centrifugation at 4,000 r.c.f. and 4 °C for 35 min in Amicon Ultra-15 filters (Millipore Ultracel-100K). Concentrated viral supernatants were stored in aliquots at -80 °C.

Before cell infection, the virus was first tested to achieve an MOI of 0.3 in H929 cells. A total of $2.1 \times 10^8$ H929 cells (70 wells per 12-well plate, 10 wells for transfection efficiency control) were transduced with the concentrated virus, attaining transduction efficiency of 25% (700 cells per lenti-CRISPRv2 construct). Puromycin was added to cells 24 h after transduction and maintained for 7 days. Subsequently, $4 \times 10^7$ cells were collected for baseline genomic DNA analysis. For the screen, cells were split into two groups of $6 \times 10^7$ transduced cells. One group was co-cultured with $6 \times 10^7$ primary NK cells (E:T ratio of 1:1) and Daratumumab (1 μg/ml). The other group was cultured under the same density and conditions but without NK cells and Daratumumab. Cells were co-cultured for 12 h, after which cells were washed with fresh media. The recovery phase was maintained for another 72 h to get 75% lysis of tumor cells.

To evaluate sgRNA enrichment, the surviving cells after co-culture or not were collected. Genomic DNA was extracted from collected tumor cells (along with the cells collected at the early time point just after puromycin selection) using the Quick-gDNA MidiPrep kit (Zymo Research, cat. no. D3100). To generate the NGS library, PCR was performed on gDNA using NEBNext® Ultra™ II Q5® Master Mix (New England Biolabs). The sequences of primers, including full barcodes, used for PCR are from Dr. Zhang's published protocol[49]. For each sample, the PCR products were pooled and then purified using the Zymo-Spin V with Reservoir. Purified libraries were quantified by Qubit dsDNA Assay Kit and sent to Novogene for quenching with 20% PhiX on Illumina Platform PE150.

To analyze the enriched sgRNA in our screen, the normalized gRNA count table was loaded into MaGeCK (Model-based Analysis of Genome-wide CRISPR-Cas9 Knockout)[50] by comparing the co-culture and control conditions described above. Top genes were determined based on mean log2 fold change (LFC) for all gRNAs and false discovery rate (FDR).

### The Cancer Genome Atlas (TCGA) correlation analysis

33 RNA-seq datasets from The Cancer Genome Atlas (TCGA) were downloaded through the R package TCGAbiolinks[51]. The gene expression in each type of cancer was normalized by DESeq2[52]. The geometric mean of gene GZMA and PRF1 in each dataset was calculated as the expression of cytolytic activity signature (CYT) first and then identified genes correlated with CYT (Pearson's $r > 0$ and $p < 0.05$). We illustrated the intersection between CYT-correlated genes and 433 CRISPR screen genes in each dataset. The Pearson's correlation coefficients of these CRISPR screen genes in each dataset were used for clustering and were displayed in the heatmap.

### CD38 sorting genome-wide CRISPR screen

The lentivirus library was the same as described above. After cell transduction and puromycin selection, 5% of cells with the lowest expression of CD38 were sorted for NGS. Data analysis is the same as described above.

### Generation of stable cell lines

For the generation of CRISPR KO cell lines, oligonucleotides (Supplementary Data 6) targeting different genes were annealed and subcloned into LentiCRISPRv2 vectors[48]. Constructs were packaged into lentivirus in HEK293T cells. Target cells were seeded in 12-well plates and spinfected with the virus for 1.5 hours at 800 x g at 35 °C, supplemented with Polybrene (8 μg/ml). The media was then aspirated, and fresh complete media was added to exclude Polybrene. After 1 day, cells were selected for stable KO using puromycin (0.5 μg/ml). After 7 days, cells were collected for immunoblotting or other experiments.

To generate complete KDM6A KO single clone cells, we delivered indicated sgRNA and cas9-GFP protein ribonucleoprotein complex into MM cells using the Neon Transfection system. One day after transfection, cells were washed, and single-cell sorted viable and GFP+

cells dispensed into 96-well cell culture plates (1 cell per well). After cells grew up, we transferred the cells to large plates and flasks. Cells were collected and KO was confirmed by immunoblotting.

### Re-introduce indicated genes into MM cells

The pLenti expression vectors were purchased from GeneCopoeia. Viral particles were packaged and concentrated as described above. MM cells were transduced with the virus and selected for stable overexpression cells using related antibiotics. After 7 days, cells were collected and identified by immunoblotting.

### Immunoblot analysis

Cells were harvested, washed with PBS, and total protein was extracted with RIPA lysis buffer supplemented with protease and phosphatase inhibitors (no. 78440, Thermo Fisher Scientific). The suspension was incubated for 15 min on ice and vortexed for 5 min. Then, samples were centrifuged at 16,000 x g at 4 °C for 10 min, and the supernatant was transferred to a new tube. Protein concentration was determined using the BCA protein assay kit (no. 23227, Thermo Fisher Scientific). Samples were mixed with 4X LDS sample buffer (no. NP0007, Thermo Fisher Scientific) and boiled at 95 °C for 8 min. Equal amounts of protein were run on NuPAGE Bis-Tris gels (Thermo Fisher Scientific) at a constant voltage and transferred to nitrocellulose membrane by iblot2 Gel Transfer Device (Thermo Fisher Scientific). Then membranes were blocked in 5% nonfat dry milk for 1 hour at room temperature, and incubated with primary Abs in 5% bovine serum albumin at 4 °C overnight. Blots were then washed three times with 1X Tris Buffered Saline with Tween (TBS-T) before incubation with secondary Abs for 1 hour. SuperSignal chemiluminescent substrate (Thermo Fisher Scientific) was used for signal detection. For reblotting the membranes, blots were stripped in stripping buffer (no. 46428, Thermo Fisher Scientific) according to the manufacturer's instruction and re-blocked.

### qRT-PCR

Total RNA was extracted with an RNeasy Mini kit (Qiagen). cDNA was generated by reverse transcription using the SuperScript VILO cDNA synthesis kit (no. 11754050, Thermo Fisher Scientific). Quantitative real-time PCR was carried out using Taqman Universal PCR master Mix and related Taqman Assay primers in a QuantStudio6 Flex real-time PCR system. The relative level of each transcript was normalized to control GAPDH expression. The primers for each gene are listed in Table S6.

### Flow cytometry

Cells were collected and washed with PBS and stained with Fixable Viability Dye eFluor 780 at 4 °C for 30 min to exclude dead cells. After the incubation, cells were washed with PBS and stained with conjugated primary antibodies or isotype control (IgG). Cells were then washed with 2% FBS containing PBS, and resuspended in Flow Staining Buffer. Cells were acquired in a BD LSRFortessa Flow Cytometer, and data were analyzed by FlowJo software.

### In vitro ADCC assay

MM cells were counted and stained with the dye Calcein Am at 37 °C for 30 min and protected from light. Then we added five times complete culture media to remove any free dye remaining in the solution. Cells were pelleted by centrifugation, resuspended in pre-warmed media, and plated in 96-well U-bottom plates (10,000 target cells per well) with 1 µg/ml Daratumumab. Following this, MM cells were thoroughly washed to remove excess antibodies before co-culturing them with effector NK cells. Natural killer cells or PBMC from healthy donors were isolated or thawed one day before the ADCC assay and added in at the indicated E:T ratio. To determine maximum and spontaneous release, target cells were mixed with an equal volume of 1% Triton-X100 in media or an equal volume of media, respectively. After incubation, plates were centrifuged at 100 x g for 3 min. Supernatants were collected to a new black 96-well plate, and the intensity of fluorescence from the supernatant was detected using 485 nm excitation and 520 nm emission. The specific killing was calculated using the following formula: 100 x (Experimental release – Spontaneous release)/ (Maximum release – Spontaneous release).

### Generation of Dara-resistant cell line

KMS11 cells were co-cultured with primary NK cells and Dara (1 µg/ml) at E:T ratio of 6:1 in 96-well U-bottom plates. Cells were incubated at 37 °C for 24 h, followed by washing and resuspending with complete media. Cells were cultured for 1 week, and the co-culture was repeated weekly 10 times (a total of 10 weeks). Resistance of cells to Dara was determined by ADCC assay.

### NK cell cytotoxicity assay

The co-culture of MM cells and primary NK cells is as described above of ADCC, but without Dara. Primary NK cells were pre-cultured with IL-2 one day before. Specific killing was calculated for ADCC.

### Intracellular interferon-γ production by NK cells

NK cells were co-cultured for 6 h at a ratio of 1:1 with MM cells in the presence or absence of Dara. After incubation, cells were washed with PBS and stained with Live/Dead Fixable dead cell stain kit (Near-IR, Thermo Fisher), followed by Fc receptor blockade. After washing, cells were stained with NK cell marker CD56-APC, fixed with Intracellular Fixation Buffer (eBioscience), permeabilized with Permeabilization Buffer (eBioscience), and stained with FITC-conjugated anti-human interferon-γ (Biolegend). Cells were acquired in a BD LSRFortessa Flow Cytometer, and analyzed by FlowJo software.

### ELISA

NK cells were co-cultured for 6 h at a ratio of 1:1 with MM cells in the presence or absence of Dara. After incubation, supernatants were collected and measured using human Granzyme B ELISA kit (Biolegend) and human Perforin ELISA kit (Abcam), according to the manufacturer's protocol.

### Human NK-reconstituted murine model of human multiple myeloma

The experimental procedures and protocol (No: 03-043) were approved by the Institutional Animal Care and Use Committee (IACUC; DFCI). All mice were housed in a pathogen-free environment at a DFCI animal facility and were handled in strict accordance with Good Animal Practice, as defined by the Office of Laboratory Animal Welfare. $2 \times 10^6$ H929-luciferase WT or KDM6A KO cells were subcutaneously injected into the right flank of 6-week-old female NOD/SCID gamma (NSG; NOD.Cg-Prkdc$^{scid}$ Il2rg$^{tm1Wjl}$/SzJ) mice lacking functional NK cells (Jackson Labs, #005557-NSG). Tumor burden was serially monitored by BLI using the IVIS Imaging System and Living Image Software (PerkinElmer, Waltham, MA, USA). After tumor engraftment was confirmed by BLI, the mice were randomly divided into 4 groups (WT + NK; WT + NK+Dara; KO + NK; KO + NK+Dara). Dara was given intraperitoneally for a total of 3 administrations (8 mg/kg per mouse weekly) starting on treatment day 1 following randomization. Human primary NK cells ($5 \times 10^6$) were injected intravenously the day after the Dara injection once per week for 3 weeks. Tumor sizes were detected by BLI once per week. The mice were euthanized by carbon dioxide before the tumors exceeded 2.0 cm in any dimension.

### ChIP-seq and ChIP-qPCR

ChIP assays were performed using a Simple Plus Enzymatic Chromatin IP kit (Magnetic Beads). Purified DNA was delivered to the molecular biology core facilities (MBCF) at Dana-Farber Cancer Institute for NGS.

The raw data were saved in files with fastq format. Low-quality reads and sequencing adapters were removed by Trimmomatic v0.38[53], and the qualified reads were mapped to the human genome (hg38) by bowtie2[54]. Peak calling was performed by MACS2[55] and visualized in IGV[56]. The intensity of peaks around the transcription start site (TSS) was calculated and shown by deeptools[57].

## RNA-seq analysis

For RNA-seq, total RNA of WT and KDM6A KO cells was extracted using the RNeasy Mini Kit (Qiagen). Library preparation and sample sequencing were performed in the molecular biology core facilities (MBCF) at Dana-Farber Cancer Institute. The RNA-Seq data were processed following the VIPER pipeline[58]. Raw reads were aligned to the human genome (hg38) by STAR[59], and the differential gene expression analysis was performed by limma[60]. The differentially expressed genes (DEG) were defined by the log of transformed fold change greater than 1 ($\log_2$(fold change)>1) and adjusted p value less than 0.05. Volcano plots were used to display the DEGs with down-regulated genes (blue) and up-regulated genes (pink). Gene set enrichment analysis (GSEA) was performed to identify significantly enriched pathways. The biologically defined gene sets were obtained from the Molecular Signatures Database (http://software.broadinstitute.org/gsea/msigdb/index.jsp). Genes used for GSEA analysis were pre-ranked on the basis of $\log_2$ fold change of TPM (transcripts per kilobase million) between WT and KDM6A KO cells.

## ATAC-seq

ATAC-seq was performed by the Genewiz Corporation. The raw reads of ATAC-seq were trimmed by Trimmomatic v0.38[53] first, and then aligned to human reference genome hg38 by bowtie2[54]. The alignment was filtered by samtools[61], screening out reads with mapping quality less than 30, inconsonant alignment, and secondary alignment. PCR duplicates and mitochondria were also filtered out prior to peak calling by Picard v2.18.26[62]. MACS2 v2.1.2[55] was used for peak calling, and the differential peak analysis was performed by R package Diffbind[63]. We summarized the peaks across the three samples and compared the intensity of peaks together. The missing peaks in any sample were assigned zero intensity. Wilcox test was used to compare KO samples and wild-type samples, and the Kruskal-Wallis test was used to examine the variance of the group.

## Statistics and reproducibility. No statistical methods were used to predetermine sample size

Two-sided student's $t$-test or analysis of variance followed by Dunnett's test is used to compare differences between the treated group and the relevant control group. Comparisons between three groups or more are done using one-way analysis of variance (ANOVA) with the Tukey post hoc test. The n number of each experiment is listed in the figure legends. Statistical tests were performed using GraphPad Prism 10.0 (GraphPad Software Inc.) unless otherwise specified. Values of $p < 0.05$(*), $p < 0.01$(**), $p < 0.001$(***), $p < 0.0001$(****) are considered significant. Values of $p > 0.05$ are indicated as ns. The repeated time of each experiment is indicated in the figure legends.

## Reporting summary

Further information on research design is available in the Nature Portfolio Reporting Summary linked to this article.

## Data availability

CRISPR screen sequencing data, RNA-seq, ChIP-seq and ATAC-seq data generated in this study are deposited in the National Institute of Health Gene Expression Omnibus (GEO) database (GSE228771). Source data are provided with this paper.

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

## Acknowledgements

We thank Molecular Biology Core Facilities, Dana-Farber Cancer Institute for assistance with RNA-seq and ChIP-seq; the animal Resources Facility, Dana-Farber Cancer Institute for support with animal studies; the flow cytometry core facility, Dana-Farber Cancer Institute for the help with cell sorting. This work was supported by the National Institutes of Health grants SPORE-P50100707 (K.C.A.), R01-CA050947 (K.C.A.), R01-CA178264 (T.H. and K.C.A.), and P01-155258 (K.C.A. and N.M.). This work was also supported by the National Natural Science Foundation of China 82200224 (L.X.) and the Natural Science Foundation of Shandong Province ZR2021MH072 (L.X.). A.G. is a Fellow of the Leukemia & Lymphoma Society and a Scholar of the American Society of Hematology; she is supported by an Individual Start-UP grant from the Italian Association for Cancer Research (AIRC) (project #27750). This study was also supported by Dr. Miriam and Sheldon G. Adelson Medical Research Foundation and the Riney Family Myeloma Initiative.

## Author contributions

Conceptualization, J.L., L.X., T.H. and K.C.A.; Methodology, J.L., L.X. and T.H.; Software, Jiang Li; Validation, J.L., L.X, N.L. and Y.L.; Formal analysis, J.L., Jiang Li, S.W. and T-Y.S.; Investigation, J.L., L.X., Jiang Li, K.W., N.L., Y.L., S.W., G.W., D.O., T-Y.S., K.K., J.P., E.M., T.W. and X.H., A.G.; Resources, Y-T.T., N.M. and P.R., R.C.; Data Curation, J.L.;

Writing-Original Draft, J.L. L.X., and T.H.; Writing-Review & Editing, all co-authors; Visualization, J.L.; Supervision, T.H. and K.C.A; Project Administration, J.L., T.H. and K.C.A; Funding Acquisition, L.X., N.M., T.H. and K.C.A.

## Competing interests

K.C.A. serves on advisory boards to Pfizer, AstraZeneca, Janssen, Starton, Window, and Bristol Myers Squibb; and is a Founder of OncoPep, C4 Therapeutics, Dynamic Cell Therapies, and NextRNA. All the other authors declare no competing interests.
