## [Peer Review File · Nature Communications]

Epigenetic regulation of CD38/CD48 by KDM6A mediates NK cell response in multiple myelomaREVIEWER COMMENTS

Reviewer #1 (Remarks to the Author):

In general, these studies are well done, novel and interesting. Other groups have evaluated mechanisms of tumor cell resistant to NK cell-mediated activity, though not in this setting. Therefore, this is an area of increasing interest. While loss of KDM6A has been demonstrated in other malignancies, the identification of KDM6A as a mediator or resistance to ab and/or cell therapy primarily through epigenetic mechanisms appears to be a new concept and may apply other areas of targeted ab and/or cell therapy. The ability to reverse this effect with an already clinically approved EZH2 inhibitor adds to the potential impact of this work. However, there are some points that could be better clarified to make this work more suitable for publication.

Major points:

1. Studies done in both Figures 1 and 7 test for myeloma cell killing using anti-CD38 abs use human NK cells combined with either daratumumab or isatuximab to test for killing of the myeloma cells. However, NK cells also typically express CD38, therefore there is a concern for NK cell-mediated fratricide that may inhibit their activity. Is this fratricide seen in these studies or is anything done to prevent it?
2. Studies done in Figure 4 use isatuximab to demonstrate the effects of resistance to ADCC is not dara-specific. However, as isa also targets CD38, these results only confirm the ADCC requires the target antigen, as expected. These studies should be done with an ab that targets a different, non-CD38 antigen, such as use of elotuzumab (elo) that targets SLAMF7. It is important to show that ADCC activity using elo is not effected by loss of KDM6A and CD38 expression. As KDM6A also leads to loss of CD48 expression, this will also help to better determine if the changes in dara-mediated activity are more due to loss of CD38 or CD48.
3. While RNAseq studies are done to evaluate changes in dara-resistant cells, additional phenotyping of the H929 and other cell lines used for KDM6A-KO studies should be evaluated by flow cytometry for typical NK cell activating and inhibitory ligands to evaluate if there may be other mechanisms that lead to this CD38-mediated ADCC. These ligands include: MICA, MICB, ULBPs, FAS-ligand, TRAIL, PVR/CD155 and HLA-class I expression.
4. Other studies have used CRISPR screens to identify novel mechanisms of tumor cell resistance to NK cells. These include work to identify DCAF15 (eLife 8:e47362) and CHMP2A (Nat Commun 13, 1899 (2022)) as potential mediators of NK cell resistance. CHMP2A is part of the ESCRTIII complex that was also shown to mediate resistance to T cell-mediated activity (Science. 2022 376(6591):377-382). Do the screens for loss of CD38-mediated ADCC demonstrate changes in any of these genes? Some discussion comparing this effect and finding of KDM6A-mediated activity to these other identified mediators of NK cell-mediated killing would be useful.

Reviewer #2 (Remarks to the Author):

This paper used genome wide CRISPR screens in HMCLs to identify the H3K27 demethylase KDM6A as a potential mediator of resistance to daratumumab-mediated ADCC. KDM6A KO was found to downregulate, via increased promoter methylation, both CD38, the antigen target of daratumumab, and CD48, a ligand that activates NK cells via the receptor 2B4. Rescue CD38 expression partially restored dara sensitivity. Rescue CD48 expression also restored dara sensitivity. Co-treatment with EZH2 inhibitors including Tazemetostat was found to restore CD38 and CD48 expression and enhance dara mediated ADCC. The data provide pre-clinical rationale for combination anti-CD38 mAb therapy with EZH2 inhibitors, supported by a small amount of clinical data suggesting that the mechanisms of resistance observed in the CRISPR screens and pre-clinical models may be operative in patients. In this way, the work is potentially significant.

By and large the data presented support the interpretations of the authors. There are a few deficits in this area. The restoration of CD38 expression in figure 2h with KDM6A rescue is not very compelling, although the corresponding extended figure 1c is more convincing. The number of clinical specimens in figures 3h+i and 5j are rather small. The work is performed in 2 classical myeloma cell lines, MM1.S and H929, which have been around for a long time, and which at this point may be less representative of myeloma biology. U266 is mentioned in the methods and would have similar issues but the data presented were generally with the other two cell lines. In practice, CD38 antibodies are now given in combination with other active myeloma agents, and EZH2 inhibitors have not yet found their way into the clinic for myeloma patients; but still, perhaps for those patients with acquired resistance to CD38 antibodies, a strategy to restore sensitivity with EZH2 inhibitors could be attempted. There are other compelling clinical strategies to treat these patients, centred around other antigen targets like BCMA etc. which might limit the clinical impact of this approach. More data from clinical specimens or even the results from treating a few patients or PDX's with this combination approach, with appropriate controls, would strengthen the impact.

Reviewer #3 (Remarks to the Author):

The manuscript by Liu et al addresses the mechanism underlying the development of resistance to treatment of multiple myeloma (MM) with anti-CD38 monoclonal antibodies (Daratumumab, Dara). Data show that indeed loss of KDM6A aligns with a decrease in CD38. This is consistent with KDM6A's function as a histone demethylase and the known role of CD38 in suppression of MM by treatment with anti-CD38 monoclonal antibodies.

Major unanswered questions however remain.

1) KDM6A affects the transcription of many genes in addition to CD38. It is unclear how much of the effect of the KDM6A KO on drug resistance is due to effects on CD38 as compared to the effect of the KDM6A KO on the numerous other genes it regulates. To answer this, one needs to determine what is the effect of KDM6A KO on Dara drug resistance in the setting of no change in CD38?

2) Since KDM6A is on the X chromosome, indeed is an X-escapee, there is an X-dosage effect, whereby it is expressed higher in females than in males. This prompts a question of physiologic significance – Do males with have lower KDM6A expression than females in the cell of focus? and do males have lower CD38 and hence more Dara resistance?

3) The paper is based almost exclusively on in vitro data of cell lines. Selective deletion of KDM6A has been done in vivo in mice to study its role in specific cell types. Experiments using mice that have a conditional KO of KDM6A in plasma cells in vivo is needed to validate in vitro findings. An in vivo experiment using male and female would add substantially to the paper's findings.

Other additional points to increase rigor of the work include:

Fig. 4 (c): Please run statistics of the result in the graph. There are only representative images now, so the reader cannot judge if there is a meaningful difference.

Fig. 7 (f and g): Please show which comparisons are statistically significant.

Extended Data Fig. 2 (b): Please run statistics, even though the difference is clear.

Extended Data Fig. 2 (c): Please run statistics. Also, please indicate what red color is.

Extended Data Fig. 3 (a and b): Please show which comparisons are statistically significant.

Extended Data Fig. 4 (a): Please show which comparisons are statistically significant. Figure legend has the explanation of asterisks, but there is no asterisk in the graph.

Extended Data Fig. 5 (a): Please run statistics.

Extended Data Fig. 6 (f and g): Please show which comparisons are statistically significant.

REVIEWER COMMENTS

Reviewer #1 (Remarks to the Author):

In general, these studies are well done, novel and interesting. Other groups have evaluated mechanisms of tumor cell resistant to NK cell-mediated activity, though not in this setting. Therefore, this is an area of increasing interest. While loss of KDM6A has been demonstrated in other malignancies, the identification of KDM6A as a mediator or resistance to ab and/or cell therapy primarily through epigenetic mechanisms appears to be a new concept and may apply other areas of targeted ab and/or cell therapy. The ability to reverse this effect with an already clinically approved EZH2 inhibitor adds to the potential impact of this work. However, there are some points that could be better clarified to make this work more suitable for publication.

Major points:

1. Studies done in both Figures 1 and 7 test for myeloma cell killing using anti-CD38 abs use human NK cells combined with either daratumumab or isatuximab to test for killing of the myeloma cells. However, NK cells also typically express CD38, therefore there is a concern for NK cell-mediated fratricide that may inhibit their activity. Is this fratricide seen in these studies or is anything done to prevent it?

Yes, we agree with the reviewer's comments. Previous research has found fratricide of NK cells in Daratumumab therapy (Wang Y *et al*, Clinical cancer research, 2018; Kararoudi MN *et al*, Blood, 2020). Consistent with these reports, we also observed NK cell fratricide in our *in vitro* assays. However, it is noteworthy that despite the occurrence of NK cell fratricide in clinical settings, Dara has demonstrated significant efficacy in myeloma patients. The complex interplay of factors in the *in vivo* environment, including the tumor microenvironment and the broader immune response, may contribute to the clinical success of anti-CD38 antibodies despite the observed fratricide phenomenon.

To address the potential impact of NK cell fratricide in our *in vitro* experiments, we pre-incubated MM cells with Dara and thoroughly washed them to remove excess antibodies before co-culturing them with effector NK cells. This strategy aims to mitigate NK cell fratricide, allowing us to assess the specific impact of anti-CD38 antibodies on myeloma cells. The methods have now been updated.

2. Studies done in Figure 4 use isatuximab to demonstrate the effects of resistance to ADCC is not dara-specific. However, as isa also targets CD38, these results only confirm the ADCC requires the target antigen, as expected. These studies should be done with an ab that targets a different, non-CD38 antigen, such as use of elotuzumab (elo) that targets SLAMF7. It is important to show that ADCC activity using elo is not effected by loss of KDM6A and CD38 expression. As KDM6A also leads to loss of CD48 expression, this will also help to better determine if the changes in dara-mediated activity are more due to loss of CD38 or CD48.

We appreciate the comment from the reviewer regarding the use of isatuximab in Figure 4 and the suggestion to further validate our findings with an antibody targeting a different antigen. We have now conducted additional experiments using elotuzumab, a monoclonal antibody that specifically targets SLAMF7, in parallel with Dara.

Prior to the ADCC assay, we performed flow cytometry analysis to confirm that the knockout of KDM6A did not alter the expression of SLAMF7. The subsequent ADCC assay results indicated that *KDM6A* KO MM cells still exhibited resistance to elotuzumab treatment. However, the observed resistance was not as pronounced as that observed with Dara. We interpret this discrepancy as arising from the fact that *KDM6A* KO leads to a reduction in the expression of both CD38 and CD48, making the cells more resistant to Dara. Conversely, SLAMF7 expression remained unaltered in *KDM6A* KO cells, suggesting that the resistance is primarily attributable to the decrease in CD48, as supported by our findings.

We have incorporated this additional data in Supplementary Fig. 4a,b to provide a more comprehensive view of the impact of *KDM6A* KO on ADCC activity with different antibodies and added it to the manuscript at Line #184. The excerpt and figure are below:

Line 184 “Overexpressing CD38 in *KDM6A*-KO MM cells only partially rescued Dara-mediated ADCC (Supplementary Fig. 3e,f), even when we utilized an anti-SLAMF7 mAb, elotuzumab, *KDM6A* KO MM cells continued to exhibit resistance to ADCC (Supplementary Fig. 4a,b). These findings imply the presence of an additional mechanism mediating ADCC resistance by *KDM6A* beyond the regulation of CD38 expression.”

3. While RNAseq studies are done to evaluate changes in dara-resistant cells, additional phenotyping of the H929 and other cell lines used for *KDM6A*-KO studies should be evaluated by flow cytometry for typical NK cell activating and inhibitory ligands to evaluate if there may be other mechanisms that lead to this CD38-mediated ADCC. These ligands include: MICA, MICB, ULBPs, FAS-ligand, TRAIL, PVR/CD155 and HLA-class I expression.

We appreciate the reviewer's suggestion to perform additional phenotyping of the H929 and other cell lines. We have now thoroughly examined our RNA-seq data, focusing on NK cell activating and inhibitory ligands in *KDM6A* KO cells. Our analysis revealed that, aside from CD48, there were no significant changes in the expression of other NK cell ligands. We also conducted flow cytometry analyses to assess the expression levels of NK cell activating and inhibitory ligands, including MICA, MICB, ULBPs, FAS-ligand, TRAIL, PVR/CD155, and HLA-class I. Our results from both RNA-seq and flow cytometry analyses consistently demonstrate that the changes induced by *KDM6A* KO primarily affect CD48 expression, and there were no significant alterations in the expression of other evaluated NK cell ligands.

The data from these analyses have been added to Supplementary Fig. 4g,h and explained in Line #202. The excerpt is below:

Line #202: “We also validated other well-studied NK activating and inhibitory ligands and found there were no significant alterations in the expression of other evaluated NK cell ligands (Supplementary Fig. 4g,h).”

id	FoldChange	p value
CD48	0.01217745	0
MICA	1.10946662	0.48358994
MICB	1.28565617	1.90E-07
HLA-A	1.03216708	0.35002269
HLA-B	0.41677866	2.54E-131
HLA-C	0.77205477	1.16E-13
ULBP2	0.35319592	0.66810733
TRAIL	0.385551	2.31E-15
CD155	0.96529767	0.31858248

4. Other studies have used CRISPR screens to identify novel mechanisms of tumor cell resistance to NK cells. These include work to identify DCAF15 (eLife 8:e47362) and CHMP2A (Nat Commun 13, 1899 (2022)) as potential mediators of NK cell resistance. CHMP2A is part of the ESCRTIII complex that was also shown to mediate resistance to T cell-mediated activity (Science. 2022 376(6591):377-382). Do the screens for loss of CD38-mediated ADCC demonstrate changes in any of these genes? Some discussion comparing this effect and finding of KDM6A-mediated activity to these other identified mediators of NK cell-mediated killing would be useful.

Yes, many CRISPR screens have been done to explore the mechanism of tumor cell resistance to NK cell treatment. Petch *et al.* found that disrupting ubiquitin ligase substrate adaptor DCAF15 sensitized a leukemia cell line to NK-mediated clearance (eLife 8:e47362, 2019). Bernareggi *et al.* using a whole-genome CRISPR-cas9 screen, discovered that CHMP2A mediates tumor cell to NK cell-mediated cytotoxicity *via* secretion of extracellular vesicles (Nat Commun 13, 1899, 2022). Recently, Sheffer *et al.* systematically studied molecular features in human tumor cells that determine their degree of sensitivity to human allogeneic NK cells using PRISM and CRISPR screens and provided a comprehensive map of mechanisms regulating tumor cell responses to NK cells (Nat Genet 53, 1196-1206, 2021). All these screens were conducted under the selective pressure of NK cells directly attacking tumor cells, while our screen was performed under ADCC selection pressure in the presence of Dara. In addition, both DCAF15 and CHMP2A were identified from depleted sgRNA screens, where their KO enhanced NK cell killing. In contrast, our primary focus was on enriched sgRNA populations. Therefore, after completing the antibiotic selection screen, we continued to culture the cells to further enrich the drug-resistant cell population.

As such, we did not observe significant changes in the expression of DCAF15 or CHMP2A. This suggests that the mechanisms underlying NK cell resistance identified in our study are distinct from those involving DCAF15 and CHMP2A. The diversity of identified mediators emphasizes the complexity of tumor cell resistance mechanisms against immune cells. While DCAF15, CHMP2A, and KDM6A contribute to resistance, their specific roles, and interactions in the context of NK cell-mediated killing might vary. Further research and comparative analyses across different mediators are crucial for a comprehensive understanding of the regulatory networks governing immune evasion in tumor cells. This could potentially uncover novel therapeutic targets and strategies for overcoming resistance in cancer immunotherapy.

We have added this in the discussion section of the manuscript at Line #275. The text is reproduced below:

Line #275: “In recent years, many CRISPR screens have been done to explore the mechanism of tumor cell resistance to NK cell treatment. Different mechanisms of resistance were found in different approaches and in different cancer types. In our study, we focused on the screens for mAb-mediated ADCC, and the mechanisms underlying NK cell resistance identified in our study are distinct from those findings. The diversity of identified mediators emphasizes the complexity of tumor cell resistance mechanisms against immune cells. Further research and comparative analyses across different mediators are crucial for a comprehensive understanding of the regulatory networks governing immune evasion in tumor cells. This could potentially uncover novel therapeutic targets and strategies for overcoming resistance in cancer immunotherapy.”

Reviewer #2 (Remarks to the Author):

This paper used genome wide CRISPR screens in HMCLs to identify the H3K27 demethylase KDM6A as a potential mediator of resistance to daratumumab-mediated ADCC. KDM6A KO was found to downregulate, via increased promoter methylation, both CD38, the antigen target of daratumumab, and CD48, a ligand that activates NK cells via the receptor 2B4. Rescue CD38 expression partially restored dara sensitivity. Rescue CD48 expression also restored dara sensitivity. Co-treatment with EZH2 inhibitors

including Tazemetostat was found to restore CD38 and CD48 expression and enhance daratumumab mediated ADCC. The data provide pre-clinical rationale for combination anti-CD38 mAb therapy with EZH2 inhibitors, supported by a small amount of clinical data suggesting that the mechanisms of resistance observed in the CRISPR screens and pre-clinical models may be operative in patients. In this way, the work is potentially significant.

By and large the data presented support the interpretations of the authors. There are a few deficits in this area. The restoration of CD38 expression in figure 2h with KDM6A rescue is not very compelling, although the corresponding extended figure 1c is more convincing. The number of clinical specimens in figures 3h+i and 5j are rather small. The work is performed in 2 classical myeloma cell lines, MM1.S and H929, which have been around for a long time, and which at this point may be less representative of myeloma biology. U266 is mentioned in the methods and would have similar issues but the data presented were generally with the other two cell lines. In practice, CD38 antibodies are now given in combination with other active myeloma agents, and EZH2 inhibitors have not yet found their way into the clinic for myeloma patients; but still, perhaps for those patients with acquired resistance to CD38 antibodies, a strategy to restore sensitivity with EZH2 inhibitors could be attempted. There are other compelling clinical strategies to treat these patients, centred around other antigen targets like BCMA etc. which might limit the clinical impact of this approach. More data from clinical specimens or even the results from treating a few patients or PDX's with this combination approach, with appropriate controls, would strengthen the impact.

We appreciate the constructive feedback provided by the reviewer. Below, we address the concerns point-by-point:

1. The restoration of CD38 expression in figure 2h with KDM6A rescue is not very compelling, although the corresponding extended figure 1c is more convincing.

We have now conducted a comprehensive analysis by assessing the mean fluorescence intensity (MFI) of CD38 in three independent rescue experiments. The results of these experiments are now presented in Supplementary Fig. 1b. We believe that the data in Supplementary Fig. 1b more convincingly demonstrates a significant increase in the surface expression of CD38 upon KDM6A overexpression, addressing the concerns raised in Figure 2h.

2. The number of clinical specimens in figures 3h+i and 5j are rather small.

We recognize the importance of sample sizes for statistical validity, so we have collected additional clinical specimens, and the corresponding q-PCR data from these specimens have been integrated into Fig. 3h,i and Fig. 5j. This increased sample size significantly enhanced the statistical power of our findings. The images are reproduced below:

3. The work is performed in 2 classical myeloma cell lines, MM1.S and H929, which have been around for a long time, and which at this point may be less representative of myeloma biology. U266 is mentioned in the methods and would have similar issues but the data presented were generally with the other two cell lines.

We acknowledge the longevity of these cell lines and recognize the continuous evolution of myeloma biology. While we understand the concern raised about the representativeness of these cell lines, including MM.1S, H929, and U266, we believe that they still serve as relevant and representative tools in preclinical studies of multiple myeloma. These cell lines have been widely utilized in the field, providing a foundation for comparative analyses, and contributing valuable insights into various aspects of myeloma biology. In addition, though we recognize the potential value of *in vivo* validation, after careful consideration and in line with the editor's guidance, we believe that is out of the scope of this paper.

4. In practice, CD38 antibodies are now given in combination with other active myeloma agents, and EZH2 inhibitors have not yet found their way into the clinic for myeloma patients; but still, perhaps for those patients with acquired resistance to CD38 antibodies, a strategy to restore sensitivity with EZH2 inhibitors could be attempted. There are other compelling clinical strategies to treat these patients, centred around other antigen targets like BCMA etc. which might limit the clinical impact of this approach.

We appreciate the insightful comments, which highlight the current clinical landscape in myeloma treatment. The combined administration of CD38 antibodies with other active agents has indeed become a standard practice, reflecting the evolving strategies to enhance therapeutic efficacy. While we recognize the importance of considering alternative clinical strategies, our study focuses on the exploration of KDM6A-mediated activity in the context of CD38-mediated ADCC resistance. Our findings contribute to the understanding of the molecular mechanisms underlying this specific aspect of resistance, which may have implications for a subset of patients with acquired resistance to CD38 antibodies. We agree that BCMA and other antigen targets offer compelling clinical strategies, and we aim to emphasize the complementary nature of our study within the broader landscape of myeloma research. Our work sheds light on a specific mechanism that may influence therapeutic response, potentially providing a rationale for exploring combination approaches, including EZH2 inhibitors, in the context of acquired resistance.

5. More data from clinical specimens or even the results from treating a few patients or PDX's with this combination approach, with appropriate controls, would strengthen the impact.

We agree with this reviewer's suggestion regarding the need for more clinical data to strengthen the impact of our study. We have collected 8 more patient samples, and the cytotoxicity data reveals that the EZH2 inhibitor increased the cytotoxicity of Daratumumab-mediated killing of multiple myeloma cells. Concurrently, there was a significant increase in CD38 and CD48 expression. These new findings contribute valuable insights into the therapeutic potential of our proposed combination approach.

We have incorporated this additional data in Fig. 7k and Supplementary Fig. 7c,d and added in an explanation at Line #252, reproduced below:

Line #252: We also confirmed that Taze increased Dara-mediated killing of MM patient cells (Fig. 7k) and increased CD38 and CD48 expression (Supplementary Fig. 7c,d).

Reviewer #3 (Remarks to the Author):

The manuscript by Liu et al addresses the mechanism underlying the development of resistance to treatment of multiple myeloma (MM) with anti-CD38 monoclonal antibodies (Daratumumab, Dara). Data show that indeed loss of KDM6A aligns with a decrease in CD38. This is consistent with KDM6A's function as a histone demethylase and the known role of CD38 in suppression of MM by treatment with anti-CD38 monoclonal antibodies.

Major unanswered questions however remain.

1) KDM6A affects the transcription of many genes in addition to CD38. It is unclear how much of the effect of the KDM6A KO on drug resistance is due to effects on CD38 as compared to the effect of the KDM6A KO on the numerous other genes it regulates. To answer this, one needs to determine what is the effect of KDM6A KO on Dara drug resistance in the setting of no change in CD38?

We appreciate the comment from the reviewer regarding the broad impact of KDM6A on the transcription of many genes beyond CD38. We acknowledge the complexity of the regulatory network influenced by *KDM6A* KO and understand the importance of discerning the specific contribution of CD38 in the context of Dara drug resistance.

We have now re-examined our RNA-seq data, which revealed numerous genes being both upregulated and downregulated by *KDM6A* KO. Recognizing CD38 as a direct target of daratumumab and its pivotal role in clinical efficacy, we have conducted additional experiments to elucidate the relationship between CD38 expression and Dara sensitivity. In Supplementary Fig. 3c,d, we overexpressed CD38 in *KDM6A* KO cells and validated the CD38 levels through flow cytometry. Surprisingly, despite higher CD38 expression in *KDM6A* KO cells compared to wild-type cells, resistance to Dara persisted, indicating that *KDM6A* KO influences Dara sensitivity through additional gene regulation beyond CD38.

Further investigation led us to identify CD48 as another KDM6A-regulated gene significantly involved in ADCC sensitivity. These findings highlight the multifaceted impact of KDM6A on drug resistance, extending beyond its effects on CD38.

2) Since KDM6A is on the X chromosome, indeed is an X-escapee, there is an X-dosage effect, whereby it is expressed higher in females than in males. This prompts a question of physiologic significance – Do males with have lower KDM6A expression than females in the cell of focus? and do males have lower CD38 and hence more Dara resistance?

We appreciate the reviewer's insightful comments regarding the potential X-dosage effect of KDM6A and its physiological significance, particularly in relation to CD38 expression and Dara resistance. Consistent with previous reports indicating that KDM6A can escape X-chromosome inactivation (Dunford, A. et al. Nature Genetics, 2017), our analysis of data from the MMRF CoMMpass study revealed a significant difference in KDM6A expression between males and females.

Intriguingly, while there was a notable difference in KDM6A expression, our analysis of CD38 expression showed no statistically significant variation between males and females. We then divided the population into groups based on KDM6A expression levels, using male average KDM6A expression as the cutoff. Our analysis unveiled a significant positive correlation between KDM6A and CD38 expression in females with low KDM6A expression, implying a role for KDM6A in positively regulating CD38 when expressed at lower levels. However, in females with high KDM6A expression, we observed no significant correlation between KDM6A and CD38, hinting at a potential saturation effect, where CD38 expression reaches a plateau at higher levels of KDM6A. This comprehensive analysis supports the concept of an X-dosage effect with KDM6A influencing CD38 expression, particularly in females with low KDM6A expression levels.

Also, in a clinical trial of Daratumumab as a monotherapy (Lancet 2016; 387: 1551-60), the overall response rate (ORR) in males was 32.7%, with 17 out of 52 patients responding, while in females, the overall response rate was 25.9%, with 14 out of 54 patients responding. That difference was not statistically significant.

3) The paper is based almost exclusively on in vitro data of cell lines. Selective deletion of KDM6A has been done in vivo in mice to study its role in specific cell types. Experiments using mice that have a conditional KO of KDM6A in plasma cells in vivo is needed to validate in vitro findings. An in vivo experiment using male and female would add substantially to the paper's findings.

We sincerely appreciate the reviewer's thoughtful suggestion for *in vivo* experiments to validate our *in vitro* findings, particularly using mice with a conditional knockout (KO) of KDM6A in plasma cells. We recognize the potential value of such *in vivo* validation. However, after careful consideration and in line with the editor's guidance, we believe that is out of the scope of this paper.

Other additional points to increase rigor of the work include:

Fig. 4 (c): Please run statistics of the result in the graph. There are only representative images now, so the reader cannot judge if there is a meaningful difference.

Thank you. We have added the statistics of the results in Supplementary Fig. 3g.

Fig. 7 (f and g): Please show which comparisons are statistically significant.

We have performed statistical analyses on the presented data, and the results are now depicted in the figure. Significantly different comparisons are indicated in the updated version.

Extended Data Fig. 2 (b): Please run statistics, even though the difference is clear.

We have conducted statistical tests on the presented data, and the results are now included in the figure.

Extended Data Fig. 2 (c): Please run statistics. Also, please indicate what red color is.

We have conducted statistical analyses for the data from three independent experiments and the results are included in the Figure. Additionally, we have provided clarification on the color representation in the figure.

Extended Data Fig. 3 (a and b): Please show which comparisons are statistically significant.

We have performed statistical analyses on the presented data. Significantly different comparisons are indicated in the figure.

Extended Data Fig. 4 (a): Please show which comparisons are statistically significant. Figure legend has the explanation of asterisks, but there is no asterisk in the graph.

We are sorry for the oversight in the figure presentation. We have conducted statistical analyses and added asterisks indicating statistical significance in Supplementary Fig. 4c.

Extended Data Fig. 5 (a): Please run statistics.

We have run the statistics, and the results are now included in the figure.

Extended Data Fig. 6 (f and g): Please show which comparisons are statistically significant.

We have run the statistical analysis and added the results in the figures.

REVIEWERS' COMMENTS

Reviewer #1 (Remarks to the Author):

The authors have done a very nice job responding to the reviewer comments and concerns. I have no additional issues.

Reviewer #2 (Remarks to the Author):

The authors have addressed all of my reviews satisfactorily.

Reviewer #3 (Remarks to the Author):

Comments have been adequately addressed.